# U-Nets as Belief Propagation: Efficient Classification, Denoising, and Diffusion in Generative Hierarchical Models

**Song Mei**
University of California, Berkeley
`songmei@berkeley.edu`

## Abstract

U-Nets are among the most widely used architectures in computer vision, renowned for their exceptional performance in applications such as image segmentation, denoising, and diffusion modeling. However, a theoretical explanation of the U-Net architecture design has not yet been fully established.

This paper introduces a novel interpretation of the U-Net architecture by studying certain generative hierarchical models, which are tree-structured graphical models extensively utilized in both language and image domains. With their encoder-decoder structure, long skip connections, and pooling and up-sampling layers, we demonstrate how U-Nets can naturally implement the belief propagation denoising algorithm in such generative hierarchical models, thereby efficiently approximating the denoising functions. This leads to an efficient sample complexity bound for learning the denoising function using U-Nets within these models. Additionally, we discuss the broader implications of these findings for diffusion models in generative hierarchical models. We also demonstrate that the conventional architecture of convolutional neural networks (ConvNets) is ideally suited for classification tasks within these models. This offers a unified view of the roles of ConvNets and U-Nets, highlighting the versatility of generative hierarchical models in modeling complex data distributions.

## 1 Introduction

U-Nets are one of the most prominent network architectures in computer vision, primarily employed for tasks such as image segmentation, denoising (Ronneberger et al., 2015; Zhou et al., 2018; Siddique et al., 2021; Oktay et al., 2018), and diffusion modeling (Sohl-Dickstein et al., 2015; Ho et al., 2020; Song & Ermon, 2019; Song et al., 2020). These networks are structured as encoder-decoder convolutional neural networks equipped with long skip connections, and their input and output typically maintain the same dimensions. While U-Nets have demonstrated exceptional performance across a variety of applications, the theoretical foundations of their key components—including the encoder-decoder structure, long skip connections, and the pooling and up-sampling layers—remain inadequately understood. Notably, long skip connections have a significant impact on performance as shown in empirical studies (Drozdzal et al., 2016; Wang et al., 2022). Existing explanations, often anecdotal, suggest their efficacy stems from improved information propagation and reduction of the vanishing gradient issue, but a thorough theoretical exploration is still lacking.

In this paper, we introduce a novel interpretation of the U-Net architecture, viewing it through the lens of neural network approximation. We posit that:

> *U-Nets naturally approximate the belief propagation denoising algorithm*
> *in certain generative hierarchical models.*

The generative hierarchical model discussed herein is a tree-structured graphical model, which has been widely employed in language and image generative modeling (Chomsky, 1959; Lee, 1996; Allen-Zhu & Li, 2023; Li et al., 2000; Willsky, 2002; Jin & Geman, 2006). We detail the precise definition of such generative hierarchical models in Section 2. A series of recent work (Mossel,

2016; Sclocchi et al., 2024; Tomasini & Wyart, 2024; Petrini et al., 2023; Kadkhodaie et al., 2023a) have pioneered the use of generative hierarchical models in studying classification tasks and diffusion models. Kadkhodaie et al. (2023a) has empirically shown that U-Nets can effectively learn the denoising function for these models. Furthermore, as noted by Sclocchi et al. (2024), the belief propagation denoising algorithm, which computes the denoising function in these models exactly, includes a downward process and an upward process, with the latter reusing the intermediate results from the downward process. In Section 4, we demonstrate how the belief propagation algorithm naturally induces the encoder-decoder structure, the long skip connections, and the pooling and up-sampling operations of the U-Nets. This gives rise to an efficient sample complexity bound for learning the denoising function in generative hierarchical models using U-Nets.

In addition to our main findings, in Section 3, we demonstrate that the standard architecture of convolutional neural networks (ConvNets) is well-suited for classification tasks within the same generative hierarchical model. We provide efficient sample complexity results to support this assertion. This offers a unified perspective on the role of both ConvNets and U-Nets in image classification and denoising tasks, and also highlights the versatility of generative hierarchical models in modeling data distributions across language and image domains.

## 2 THE GENERATIVE HIERARCHICAL MODEL

To define the generative hierarchical model, we start by introducing some key notations. Consider a tree $\mathcal{T} = (\mathcal{V}, \mathcal{E})$ with a height of $L$, where we conventionally designate the root of the tree r as layer 0. For each node $v \in \mathcal{V}$, we denote $\mathrm{pa}(v)$ as the parent of $v$, $\mathcal{C}(v)$ as the children of $v$, and $\mathcal{N}(v)$ as the siblings of $v$. We denote $\mathcal{V}^{(\ell)}$ as the set of nodes at layer $\ell$. We assume that for any $v \in \mathcal{V}^{(\ell-1)}$, the number of children is precisely $m^{(\ell)}$ for $\ell \in [L]$. The leaf nodes $v \in \mathcal{V}^{(L)}$ have no children. Additionally, for each $v \in \mathcal{V}^{(\ell)}$, we assume an ordering function (a bijection) $\iota : \mathcal{C}(v) \to [m^{(\ell)}]$, ensuring that any child $v' \in \mathcal{C}(v)$ possesses a unique rank $\iota(v') \in [m^{(\ell)}]$. We denote the number of nodes at layer $\ell$ as $d^{(\ell)}$, and the number of nodes at layer $L$ as $d = d^{(L)}$. We further denote $\underline{m} = (m^{(\ell)})_{\ell \in [L]}$, and $\|\underline{m}\|_1 = \sum_{\ell=1}^{L} m^{(\ell)}$. By these definitions and assumptions, we have $d^{(\ell)} = \prod_{1 \le s \le \ell} m^{(s)}$, and $1 = d^{(0)} \le d^{(1)} \le \cdots \le d^{(L)} = d$.

For each layer $\ell = 0, \ldots, L$, every tree node $v \in \mathcal{V}^{(\ell)}$ is associated with a variable $x_v^{(\ell)} \in [S]$ for some $S \in \mathbb{N}_{\ge 2}$. (For simplicity, we use the same variable space $[S]$ across all layers, although our framework can accommodate variations across different layers $\ell$.) We denote $\boldsymbol{x}^{(\ell)} = (x_v^{(\ell)})_{v \in \mathcal{V}^{(\ell)}} \in [S]^{d^{(\ell)}}$ as the variables at layer $\ell$. The variables at the leaves, $\boldsymbol{x} = \boldsymbol{x}^{(L)} \in [S]^d$ are considered the observed covariates, exemplified by the pixel representation of an image. Conversely, the root node variable $y = x_{\mathrm{r}}^{(0)} \in [S]$ is treated as the associated label. Variables for the intermediate layers $\{\boldsymbol{x}^{(\ell)}\}_{1 \le \ell \le L-1}$ remain unobserved.

**The generative hierarchical model.** We consider a specific type of generative hierarchical model (GHM)[1] , which is a joint distribution $\mu_\star$ over variables

$$(y = x^{(0)} \in [S], \quad \boldsymbol{x}^{(1)} \in [S]^{d^{(1)}}, \quad \ldots, \quad \boldsymbol{x}^{(L-1)} \in [S]^{d^{(L-1)}}, \quad \boldsymbol{x}^{(L)} = \boldsymbol{x} \in [S]^d),$$

associated with a set of functions $\{\psi^{(\ell)} : [S] \times [S]^{m^{(\ell)}} \to \mathbb{R}_{\ge 0}\}_{\ell \in [L]}$, defined as

$$\begin{aligned} &\mu_\star(y, \boldsymbol{x}^{(1)}, \ldots, \boldsymbol{x}^{(L-1)}, \boldsymbol{x}) \\ &\propto \psi^{(1)}(y, \boldsymbol{x}^{(1)}) \cdot \Big( \textstyle\prod_{v \in \mathcal{V}^{(1)}} \psi^{(2)}(x_v^{(1)}, x_{\mathcal{C}(v)}^{(2)}) \Big) \cdots \Big( \textstyle\prod_{v \in \mathcal{V}^{(L-1)}} \psi^{(L)}(x_v^{(L-1)}, x_{\mathcal{C}(v)}) \Big). \end{aligned} \tag{GHM}$$

The formula specifies that any two nodes $v_1, v_2 \in \mathcal{V}^{(\ell)}$ within the same level uses the same function $\psi^{(\ell)}$, thereby embedding specific invariance properties into $\mu_\star$. Consequently, this GHM ensures that $(x_v)_{v \succeq v_1} \overset{d}{=} (x_v)_{v \succeq v_2}$ for any $v_1, v_2 \in \mathcal{V}^{(\ell)}$. The notation $v \succeq v_1$ denotes that $v$ is either

---

[1] We define "generative hierarchical models" as general probabilistic models with a hierarchical structure. The specific model discussed in this paper is an instance of such generative hierarchical models. These models are also known by various other names, including "hierarchical generative models", "latent hierarchical models", "Bayesian hierarchical models", "hierarchical Markov random fields", among others.

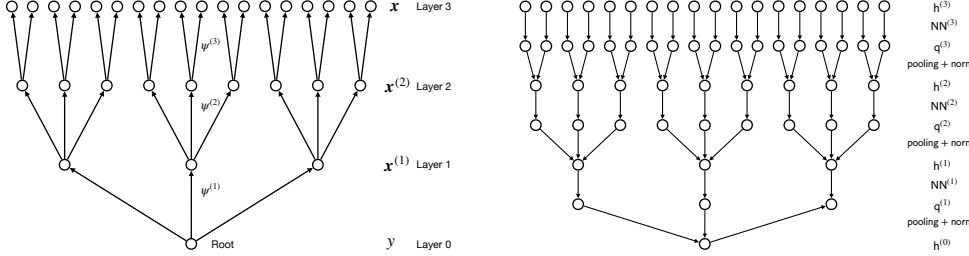

Figure 1: Left: The generative hierarchical model with 3 layers and $m^{(1)} = 3$, $m^{(2)} = 3$, and $m^{(3)} = 2$ children in each layer. Right: A 3-layer convolutional neural network.

identical to or a descendant of $v_1$. We will explore later how this invariance property interacts with the convolutional structure of the neural networks to be introduced.

Throughout the paper, we impose a specific assumption on the $\psi$ functions in GHM, namely that each $\psi$ function can be factorized into a product of functions that depend solely on the ordering of the child. This assumption is essential for enabling convolutional neural networks to approximate the associated belief propagation algorithm.

**Assumption 1** (Factorization of $\psi$). *For each layer $\ell \in [L]$ and node $v \in \mathcal{V}^{(\ell-1)}$, we have*

$$\psi^{(\ell)}(x_v^{(\ell-1)}, x_{\mathcal{C}(v)}^{(\ell)}) = \prod_{v' \in \mathcal{C}(v)} \psi_{\iota(v')}^{(\ell)}(x_v^{(\ell-1)}, x_{v'}^{(\ell)}). \tag{1}$$

We also state a technical assumption concerning the boundedness of these $\psi$ functions of GHM.

**Assumption 2** (Boundedness of $\psi$). *For any layer $\ell \in [L]$ and child node rank $\iota \in [m^{(\ell)}]$, the transition probabilities are bounded as follows:*

$$1/K \leq \min_{x,x'} \psi_\iota^{(\ell)}(x, x') \leq \max_{x,x'} \psi_\iota^{(\ell)}(x, x') \leq K. \tag{2}$$

It is helpful to think about the joint distribution $\mu_\star$ as a tree-structured Markov process, which admits the factorization

$$\begin{aligned}
&\mu_\star(y, \boldsymbol{x}^{(1)}, \ldots, \boldsymbol{x}^{(L-1)}, \boldsymbol{x}) \\
&= \mu_\star(y)\mu_\star(\boldsymbol{x}^{(1)}|y) \cdot \left( \prod_{v \in \mathcal{V}^{(1)}} \mu_\star(x_{\mathcal{C}(v)}^{(2)}|x_v^{(1)}) \right) \cdots \left( \prod_{v \in \mathcal{V}^{(L-1)}} \mu_\star(x_{\mathcal{C}(v)}|x_v^{(L-1)}) \right),
\end{aligned} \tag{3}$$

where we abuse the notation to denote $\mu_\star(x_{\mathcal{C}(v)}^{(\ell)}|x_v^{(\ell-1)})$ as the conditional probability of $x_{\mathcal{C}(v)}^{(\ell)}$ given $x_v^{(\ell-1)}$, and $\mu_\star(y)$ as the marginal probability of $y$. Indeed, any graphical model specified as in Eq. (1) can be cast into the form of (3). Furthermore, It can be checked that Eq. (3) coincides with (GHM) upon taking $\psi^{(\ell)}(x_v^{(\ell-1)}, x_{\mathcal{C}(v)}^{(\ell)}) = \mu_\star(x_{\mathcal{C}(v)}^{(\ell)}|x_v^{(\ell-1)})$ for $\ell \geq 1$, and $\psi^{(1)}(y, \boldsymbol{x}^{(1)}) = \mu_\star(y)\mu_\star(\boldsymbol{x}^{(1)}|y)$. In this scenario, the factorization assumption (1) implies that $(x_{v'}^\ell)_{v' \in \mathcal{C}(v)}$ given $x_v^{(\ell-1)}$ are conditionally independent. We include a schematic plot of the generative hierarchical model with 3 layers as in Figure 1(left).

**GHMs as natural models for languages and images.** In the field of linguistics, GHMs are very similar to context-free grammars (CFGs) (Chomsky, 1959; Lee, 1996; Allen-Zhu & Li, 2023). The generative process of a context-free grammar involves creating valid strings or sentences based on a given set of production rules (the $\psi$ functions) that dictate how symbols can be extended to form new strings. Starting with an initial symbol (the label $y$), the generation proceeds iteratively by applying production rules until all symbols belong to the terminal set, thus forming a complete sentence (the covariate $\boldsymbol{x}$).

In computer vision, GHMs are often utilized to model natural images (Li et al., 2000; Willsky, 2002; Jin & Geman, 2006), where they are sometimes referred to as multi-resolution Markov models. The hypothetical image generation process begins with a high-level concept of the image (represented

by the label $y$), which is then iteratively refined using a production rule (the $\psi$ function). This refinement continues through successive resolution levels until the detail reaches the pixel level, resulting in the final image (the covariate $\boldsymbol{x}$).

We remark that a series of studies (Sclocchi et al., 2024; Tomasini & Wyart, 2024; Petrini et al., 2023; Kadkhodaie et al., 2023a) have provided both theoretical and empirical evidence supporting the efficacy of GHMs as powerful tools for modeling the properties of combinatorial data.

## 3 THE WARM-UP PROBLEM: CLASSIFICATION IN GHMS

In this section, we consider the warm-up problem of classification within the GHM. In the subsequent section, we will investigate the denoising task, where the results and intuitions will be similar and parallel to those presented in this section.

In the classification task, consider the scenario where we observe a set of iid samples $\{(\boldsymbol{x}_i, y_i)\}_{i \in [n]}$ drawn from $\mu_\star$ under the GHM. Our objective is to learn a probabilistic classifier $\hat{\mu}(y|\boldsymbol{x})$ from this dataset. With a suitable loss function, the optimal classifier is the Bayes classifier $\mu_\star(y|\boldsymbol{x})$, which represents the true conditional probability of $y$ given $\boldsymbol{x}$. We aim to examine the sample complexity of learning this classifier through empirical risk minimization over the class of convolutional networks.

**The ConvNet architecture.** We here introduce the convolutional neural network (ConvNet) architecture used for classification, represented as $\mu_{\mathrm{NN}}(\cdot|\boldsymbol{x}) \in \Delta([S])$ for input $\boldsymbol{x} \in [S]^d$. Initially, we set $\mathsf{h}_v^{(L)} = x_v \in [S]$ for each node $v \in \mathcal{V}^{(L)}$. The operational flow of the network unfolds as follows:

$$\mathsf{q}_v^{(\ell)} = \mathrm{NN}_{\iota(v)}^{(\ell)}(\mathsf{h}_v^{(\ell)}) \in \mathbb{R}^S, \qquad\qquad \ell \in [L], \ \ v \in \mathcal{V}^{(\ell)},$$

$$\mathsf{h}_v^{(\ell-1)} = \mathrm{normalize}\Big( \sum_{v' \in \mathcal{C}(v)} \mathsf{q}_{v'}^{(\ell)} \Big) \in \mathbb{R}^S, \quad \ell \in [L], \ \ v \in \mathcal{V}^{(\ell-1)}, \qquad \text{(ConvNet)}$$

$$\mu_{\mathrm{NN}}(\cdot|\boldsymbol{x}) = \mathrm{softmax}(\mathsf{h}_{\mathrm{r}}^{(0)}) \in \Delta([S]).$$

The functions $\mathrm{NN}_{\iota(v)}^{(\ell)} : \mathbb{R}^{S_{\mathrm{in}}^{(\ell)}} \to \mathbb{R}^S$ (which adjust their input dimensionality $S_{\mathrm{in}}^{(\ell)}$ based on layer depth, $S_{\mathrm{in}}^{(\ell)} = (S+1) \cdot \mathbf{1}\{\ell \neq L\} + 2 \cdot \mathbf{1}\{\ell = L\}$) and $\mathrm{normalize} : \mathbb{R}^S \to \mathbb{R}^S$ are defined as follows:

$$\mathrm{NN}_\iota^{(\ell)}(\mathsf{h}) = W_{1,\iota}^{(\ell)} \cdot \mathrm{ReLU}(W_{2,\iota}^{(\ell)} \cdot \mathrm{ReLU}(W_{3,\iota}^{(\ell)} \cdot [\mathsf{h}; 1])), \tag{4}$$

$$\mathrm{normalize}(h)_s = h_s - \max_{s' \in [S]} h_{s'}. \tag{5}$$

The dimensions of the weight matrices within the ConvNet are specified as follows:

$$\boldsymbol{W} = \Big\{ \{W_{1,\iota}^{(\ell)} \in \mathbb{R}^{S \times D}\}_{\iota \in [m^{(\ell)}]}, \{W_{2,\iota}^{(\ell)} \in \mathbb{R}^{D \times D}\}_{\iota \in [m^{(\ell)}]}, \{W_{3,\iota}^{(\ell)} \in \mathbb{R}^{D \times S_{\mathrm{in}}^{(\ell)}}\}_{\iota \in [m^{(\ell)}]} \Big\}_{\ell \in [L]}. \tag{6}$$

Furthermore, we denote $\mu_{\mathrm{NN}}^{\boldsymbol{W}}$ as the ConvNet classifier $\mu_{\mathrm{NN}}$ parameterized by $\boldsymbol{W}$. A schematic illustration of a ConvNet with 3 layers is provided in Figure 1(right).

**Remark 1** (An explanation of the "ConvNet" architecture). *We remark that the neural network layers described in (ConvNet) are different from the "convolution operations" typically seen in practice. The convolution operations used in practice involve computing the inner products between convolutional filters and image patches, whereas in (ConvNet) and (4), a point-wise product is employed instead. Despite this, these layers are still referred to as convolutional layers because the mapping from $\{\mathsf{h}_v^{(\ell)}\}_{v \in \mathcal{V}^{(\ell)}}$ to $\{\mathsf{q}_v^{(\ell)}\}_{v \in \mathcal{V}^{(\ell)}}$, as per the first line of (ConvNet), preserves the translation-invariance property. Specifically, we use the same function $\mathrm{NN}_\iota^{(\ell)}$ across different inputs $\mathsf{h}_{v_1}^{(\ell)}$ and $\mathsf{h}_{v_2}^{(\ell)}$ as long as $\iota(v_1) = \iota(v_2) = \iota$.*

*Additionally, the "normalization operator" defined in (5) differs from commonly used ones. We adopt this specific form for technical reasons, to effectively control the approximation error.*

*Despite these differences from standard convolutional networks, (ConvNet) represents an iterative composition of convolutional layers, pooling layers, and normalization layers, aligning closely with the architecture of convolutional networks used in practice. Figure 1(right) shows the sequence of these operations in detail.*

**The ERM estimator.** In the classification task, we employ empirical risk minimization over ConvNets as outlined in the following equation:

$$\widehat{\boldsymbol{W}} = \arg \min_{\boldsymbol{W} \in \mathcal{W}_{d,\underline{m},L,S,D,B}} \left\{ \widehat{\mathsf{R}}(\mu_{\mathrm{NN}}^{\boldsymbol{W}}) = \frac{1}{n} \sum_{i=1}^{n} \mathrm{loss}(y_i, \mu_{\mathrm{NN}}^{\boldsymbol{W}}(\cdot|\boldsymbol{x}_i)) \right\}, \tag{7}$$

where, for simplicity of analysis, we opt for the square loss rather than the more commonly used cross-entropy loss:

$$\mathrm{loss}(y, \mu_{\mathrm{NN}}^{\boldsymbol{W}}(\cdot|\boldsymbol{x})) = \sum_{s=1}^{S} \left( 1\{y = s\} - \mu_{\mathrm{NN}}^{\boldsymbol{W}}(s|\boldsymbol{x}) \right)^2. \tag{8}$$

The parameter space for the ConvNets is defined as:

$$\mathcal{W}_{d,\underline{m},L,S,D,B} := \left\{ \boldsymbol{W} \text{ as defined in (6)} : \|\boldsymbol{W}\| := \max_{j \in [3]} \max_{\ell \in [L]} \max_{\iota \in [m^{(\ell)}]} \|W_{j,\iota}^{(\ell)}\|_{\mathrm{op}} \leq B \right\}. \tag{9}$$

We anticipate that the empirical risk minimizer, $\mu_{\mathrm{NN}}^{\widehat{\boldsymbol{W}}}$, could learn the Bayes classifier $\mu_\star$, as the global minimizer of the population risk over all conditional distributions yields the Bayes classifier:

$$\mu_\star(\cdot|\cdot) = \arg \min_\mu \left\{ \mathsf{R}(\mu) = \mathbb{E}[\mathrm{loss}(y, \mu(\cdot|\boldsymbol{x}))] \right\}.$$

In our theoretical analysis, we measure the discrepancy between $\mu_{\mathrm{NN}}^{\widehat{\boldsymbol{W}}}$ and $\mu_\star$ using the squared Euclidean distance:

$$\mathsf{D}_2^2(\mu, \mu_\star) = \mathbb{E}_{\boldsymbol{x} \sim \mu_\star} \left[ \sum_{s=1}^{S} \left( \mu(s|\boldsymbol{x}) - \mu_\star(s|\boldsymbol{x}) \right)^2 \right]. \tag{10}$$

**Sample complexity bound.** The subsequent theorem establishes the bound of the $\mathsf{D}_2^2$-distance between the ConvNet estimator $\mu_{\mathrm{NN}}^{\widehat{\boldsymbol{W}}}$ and the true Bayes classifier $\mu_\star$.

**Theorem 1** (Learning to classify using ConvNets). *Let Assumption 1 and 2 hold. Let $\mathcal{W}_{d,\underline{m},L,S,D,B}$ be the set defined as in Eq. (9), where $D \geq S^2 K^2 d \cdot 3^L$ and $B = \mathrm{Poly}(d, S, K, 3^L, D)$. Let $\widehat{\boldsymbol{W}}$ be the empirical risk minimizer as in Eq. (7). Then with probability at least $1 - \eta$, we have*

$$\mathsf{D}_2^2(\mu_{\mathrm{NN}}^{\widehat{\boldsymbol{W}}}, \mu_\star) \leq C \cdot \left( \frac{S^4 K^4 d^2 \cdot 3^{2L}}{D^2} + \sqrt{\frac{LD(D + 2S + 1)\|\underline{m}\|_1 \log(d\|\underline{m}\|_1 DS \cdot 3^L) + \log(1/\eta)}{n}} \right). \tag{11}$$

The proof of Theorem 1 is detailed in Section E.

**Remark 2.** *To ensure the $\mathsf{D}_2^2$-distance is less than $\epsilon^2$, Theorem 1 requires to take*

$$D = \Theta(S^2 K^2 d 3^L / \epsilon), \quad n = \tilde{\Theta}(L S^4 K^4 d^2 3^{2L} \|\underline{m}\|_1 / \epsilon^6), \tag{12}$$

*where $\tilde{\Theta}$ hides a logarithmic factor $\log(d\|\underline{m}\|_1 S \cdot 3^L / (\eta\epsilon))$. The dependency on any of the parameters $(S, d, 3^L, K, \|\underline{m}\|_1, \epsilon)$ could potentially be refined by imposing additional assumptions on the $\psi$ functions or through a more detailed analysis of approximation and generalization. This question of improving rates remains open for future work.*

*Consider a simplified scenario where $S = 2$, $K$ is constant, and $m^{(\ell)} = m \geq 3$ for each $\ell \in [L]$, leading to $d = m^L$. In this setup, the sample complexity gives*

$$n = \tilde{\Theta}(L^2 m \cdot d^{2+2\log_m 3} / \epsilon^6) \leq \tilde{\Theta}(L^2 m \cdot d^4 / \epsilon^6), \tag{13}$$

*exhibiting a polynomial dependence on $d$ and $1/\epsilon$. Such polynomial scaling aligns with existing literature (Poggio et al., 2017; Malach & Shalev-Shwartz, 2020; Schmidt-Hieber, 2020; Allen-Zhu & Li, 2022; Petrini et al., 2023), which indicates that learning hierarchical models using multi-layer networks avoids the curse of dimensionality. Theorem 1 serves as a warm-up result in the classification context. In Section 4, we aim to extend similar methodologies to address denoising problems, employing analogous proof strategies.*

**Proof strategy.** The proof strategy begins by decomposing the squared distance between the learned model and the true Bayes classifier into approximation and generalization error terms. The generalization error is bounded using a standard chaining argument, leading to a rate of $\tilde{O}(\sqrt{d_{\mathrm{p}}/n})$, where $d_{\mathrm{p}}$ denotes the number of ConvNet parameters. The focus then shifts to controlling the approximation error. This is done by first introducing the belief propagation and message passing algorithm for computing the Bayes classifier, and subsequently showing that ConvNets can approximate this algorithm effectively. A detailed outline of the proof strategy is provided in Appendix A.1, with the complete proof presented in Appendix E.

## 4 DENOISING AND DIFFUSION IN GHMS

In this section, we consider the denoising task within the GHM. Consider the joint distribution of noisy and clean covariates $(\boldsymbol{z}, \boldsymbol{x})$, generated from the following: $\boldsymbol{x} \sim \mu_\star$ represents the clean covariates, and $\boldsymbol{z} = \boldsymbol{x} + \boldsymbol{g}$ where $\boldsymbol{g} \sim \mathcal{N}(\boldsymbol{0}, \mathbf{I}_d)$ denotes the independent isotropic Gaussian noise. For simplicity in notation and with a slight abuse of notations, we continue to refer to the joint distribution of $(\boldsymbol{z}, \boldsymbol{x})$ as $\mu_\star$.

We consider a scenario where a set of iid samples $\{(\boldsymbol{z}_i, \boldsymbol{x}_i)\}_{i \in [n]} \sim_{iid} \mu_\star$ is drawn from the distribution. Our objective is to learn a denoiser $\boldsymbol{m}(\boldsymbol{z})$ from this dataset. With a suitable loss function, the optimal denoiser is the Bayes denoiser $\boldsymbol{m}_\star(\boldsymbol{z}) = \mathbb{E}_{(\boldsymbol{x}, \boldsymbol{z}) \sim \mu_\star}[\boldsymbol{x}|\boldsymbol{z}]$, which calculates the posterior expectation of $\boldsymbol{x}$ given $\boldsymbol{z}$. We aim to examine the sample complexity of learning this denoiser through empirical risk minimization over the class of U-Nets. The approaches and results of this section closely align with those discussed in Section 3 on the classification task.

**The U-Net architecture.** We here introduce the U-Net architecture used for denoising, represented as $\boldsymbol{m}_{\mathrm{NN}}(\boldsymbol{z}) \in \mathbb{R}^d$ for input $\boldsymbol{z} \in \mathbb{R}^d$. Initially, we set $\mathsf{h}_{\downarrow,v}^{(L)} = (-(x - z_v)^2/2)_{x \in [S]} \in \mathbb{R}^S$ for $v \in \mathcal{V}^{(L)}$ for each node $v \in \mathcal{V}^{(L)}$. The operational flow of the network unfolds as follows:

$$
\begin{aligned}
\mathsf{q}_{\downarrow,v}^{(\ell)} &= \mathrm{NN}_{\downarrow,\iota(v)}^{(\ell)}(\mathrm{normalize}(\mathsf{h}_{\downarrow,v}^{(\ell)})) \in \mathbb{R}^S, & \ell \in [L], \quad v \in \mathcal{V}^{(\ell)}, \\
\mathsf{h}_{\downarrow,v}^{(\ell-1)} &= \textstyle\sum_{v' \in \mathcal{C}(v)} \mathsf{q}_{\downarrow,v'}^{(\ell)} \in \mathbb{R}^S, & \ell \in [L], \quad v \in \mathcal{V}^{(\ell-1)}, \\
\mathsf{u}_{\uparrow,v}^{(\ell)} &= \mathsf{b}_{\uparrow,\mathrm{pa}(v)}^{(\ell-1)} \in \mathbb{R}^S, \quad (\text{with } \mathsf{b}_{\uparrow,\mathrm{r}}^{(0)} = \mathsf{h}_{\downarrow,\mathrm{r}}^{(0)}) & \ell \in [L], \quad v \in \mathcal{V}^{(\ell)}, \qquad \text{(UNet)} \\
\mathsf{b}_{\uparrow,v}^{(\ell)} &= \mathrm{NN}_{\uparrow,\iota(v)}^{(\ell)}(\mathrm{normalize}(\mathsf{u}_{\uparrow,v}^{(\ell)} - \mathsf{q}_{\downarrow,v}^{(\ell)})) + \mathsf{h}_{\downarrow,v}^{(\ell)} \in \mathbb{R}^S, & \ell \in [L], \quad v \in \mathcal{V}^{(\ell)}, \\
\boldsymbol{m}_{\mathrm{NN}}(\boldsymbol{z})_v &= \textstyle\sum_{s \in [S]} s \cdot \mathrm{softmax}(\mathsf{b}_{\uparrow,v}^{(L)})_s, & v \in \mathcal{V}^{(L)}.
\end{aligned}
$$

The functions $\{\mathrm{NN}_{\downarrow,\iota}^{(\ell)}, \mathrm{NN}_{\uparrow,\iota}^{(\ell)} : \mathbb{R}^S \to \mathbb{R}^S\}_{\ell \in [L]}$ and normalize $: \mathbb{R}^S \to \mathbb{R}^S$ are defined as follows:

$$
\mathrm{NN}_{\diamond,\iota}^{(\ell)}(\mathsf{h})_s = W_{1,\diamond,\iota}^{(\ell)} \cdot \mathrm{ReLU}(W_{2,\diamond,\iota}^{(\ell)} \cdot \mathrm{ReLU}(W_{3,\diamond,\iota}^{(\ell)} \cdot [\mathsf{h}; 1])), \quad s \in [S], \diamond \in \{\downarrow, \uparrow\}, \ell \in [L], \iota \in [m^{(\ell)}], \tag{14}
$$

$$
\mathrm{normalize}(h)_s = h_s - \max_{s' \in [S]} h_{s'}. \tag{15}
$$

The dimensions of the weight matrices within the U-Net are specified as follows:

$$
\boldsymbol{W} = \left\{ \{W_{1,\diamond,\iota}^{(\ell)} \in \mathbb{R}^{S \times D}\}_{\iota \in [m^{(\ell)}]}, \{W_{2,\diamond,\iota}^{(\ell)} \in \mathbb{R}^{D \times D}\}_{\iota \in [m^{(\ell)}]}, \{W_{3,\diamond,\iota}^{(\ell)} \in \mathbb{R}^{D \times (S+1)}\}_{\iota \in [m^{(\ell)}]} \right\}_{\ell \in [L], \diamond \in \{\downarrow, \uparrow\}} \tag{16}
$$

Furthermore, we denote $\boldsymbol{m}_{\mathrm{NN}}^{\boldsymbol{W}}$ as the U-Net denoiser $\boldsymbol{m}_{\mathrm{NN}}$ parameterized by $\boldsymbol{W}$. A schematic illustration of a U-Net with 3 layers is provided in Figure 2.

**Remark 3** (An explanation of the "U-Net" architecture). *As noted in our discussion of the convolutional network as in Remark 1, the "convolutional layers" and the "normalization operator" described in (UNet) are different from practical implementations. However, we continue to use these terms because they retain core characteristics of their practical counterparts.*

*An important feature of (UNet) is its encoder-decoder architecture and the inclusion of long skip connections, which closely mirror practical implementations. Specifically, the encoder sequence in (UNet) progresses as $\mathsf{h}_\downarrow^{(L)} \to \mathsf{q}_\downarrow^{(L)} \to \mathsf{h}_\downarrow^{(L-1)} \to \cdots \to \mathsf{q}_\downarrow^{(1)} \to \mathsf{h}_\downarrow^{(0)}$, consisting of a series*

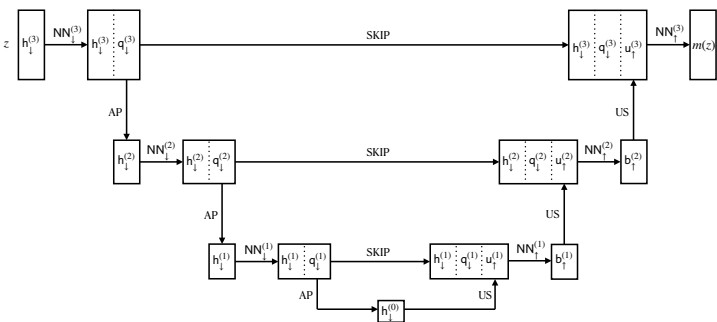

Figure 2: A U-Net with $L = 3$. "AP" stands for average-pooling. "US" stands for up-sampling. "SKIP" stands for long skip connections.

*of convolutional, average pooling, and normalization layers. This part of the architecture is the same as (ConvNet) for the classification task. The decoder sequence ascends as $\mathsf{b}_\uparrow^{(0)} \to \mathsf{u}_\uparrow^{(1)} \to \mathsf{b}_\uparrow^{(1)} \to \cdots \to \mathsf{b}_\uparrow^{(L-1)} \to \mathsf{u}_\uparrow^{(L)} \to \boldsymbol{m}_{\mathrm{NN}}$, consisting of a series of convolutional, up-sampling, and normalization layers. Moreover, the computation of $\mathsf{b}_\uparrow^{(\ell)}$ utilizes $\mathsf{u}_\uparrow^{(\ell)}$ from the upward sequence, and $(\mathsf{h}_\downarrow^{(\ell)}, \mathsf{q}_\downarrow^{(\ell)})$ from the downward process, which is enabled by the long skip connections. This encoder-decoder architecture and the long skip connections are effectively visualized in Figure 2.*

**The ERM estimator.**    In the denoising task, we employ empirical risk minimization over U-Nets as outlined in the following equation:

$$\widehat{\boldsymbol{W}} = \arg\min_{\boldsymbol{W} \in \mathcal{W}_{d,\underline{m},L,S,D,B}} \left\{ \widehat{\mathsf{R}}(\boldsymbol{m}_{\mathrm{NN}}^{\boldsymbol{W}}) = \frac{1}{nd} \sum_{i=1}^{n} \|\boldsymbol{x}_i - \boldsymbol{m}_{\mathrm{NN}}^{\boldsymbol{W}}(\boldsymbol{z}_i)\|_2^2 \right\}. \tag{17}$$

The parameter space for the U-Nets is defined as:

$$\mathcal{W}_{d,\underline{m},L,S,D,B} := \left\{ \boldsymbol{W} \text{ as defined in (16)} : \|\boldsymbol{W}\| := \max_{\diamond \in \{\downarrow,\uparrow\}} \max_{j \in [3]} \max_{\ell \in [L]} \max_{\iota \in [m^{(\ell)}]} \|W_{j,\iota}^{(\ell)}\|_{\mathrm{op}} \leq B \right\}. \tag{18}$$

We anticipate that the empirical risk minimizer, $\boldsymbol{m}_{\mathrm{NN}}^{\widehat{\boldsymbol{W}}}$, could learn the Bayes denoiser $\boldsymbol{m}_\star$, as the global minimizer of the population risk over all functions yields the Bayes denoiser:

$$\boldsymbol{m}_\star(\cdot) = \arg\min_{\boldsymbol{m}} \left\{ \mathsf{R}(\boldsymbol{m}) = \mathbb{E}_{(\boldsymbol{x},\boldsymbol{z}) \sim \mu_\star}[d^{-1}\|\boldsymbol{x} - \boldsymbol{m}(\boldsymbol{z})\|_2^2] \right\}.$$

In our theoretical analysis, we measure the discrepancy between $\boldsymbol{m}_{\mathrm{NN}}^{\widehat{\boldsymbol{W}}}$ and $\boldsymbol{m}_\star$ using the squared Euclidean distance:

$$\mathsf{D}_2^2(\boldsymbol{m}, \boldsymbol{m}_\star) = \mathbb{E}_{\boldsymbol{z} \sim \mu_\star}\left[ d^{-1}\|\boldsymbol{m} - \boldsymbol{m}_\star(\boldsymbol{z})\|_2^2 \right]. \tag{19}$$

**Sample complexity bound.**    The subsequent theorem establishes the bound of the $\mathsf{D}_2^2$-distance between the U-Net estimator $\boldsymbol{m}_{\mathrm{NN}}^{\widehat{\boldsymbol{W}}}$ and the true Bayes denoiser $\boldsymbol{m}_\star$.

**Theorem 2** (Learning to denoise using U-Nets). *Let Assumption 1 and 2 hold. Let $\mathcal{W}_{d,\underline{m},L,S,D,B}$ be the set defined as in Eq. (18), where $B = \mathrm{Poly}(d, S, K, 18^L, D)$. Let $\widehat{\boldsymbol{W}}$ be the empirical risk minimizer as in Eq. (17). Then with probability at least $1 - \eta$, we have*

$$\mathsf{D}_2^2(\boldsymbol{m}_{\mathrm{NN}}^{\widehat{\boldsymbol{W}}}, \boldsymbol{m}_\star) \leq C \cdot \left( \frac{S^6 K^4 d^2 \cdot 18^{2L}}{D^2} + S^2 \cdot \sqrt{\frac{LD(D+2S+1)\|\underline{m}\|_1 \log(d\|\underline{m}\|_1 DS \cdot 18^L) + \log(1/\eta)}{n}} \right).$$

The proof of Theorem 2 is detailed in Section H.

**Remark 4.** *To ensure the* $\mathsf{D}_2^2$*-distance is less than* $\epsilon^2$*, Theorem 2 requires to take*

$$D = \Theta(S^3 K^2 d 18^L/\epsilon), \quad n = \tilde{\Theta}(LS^{10}K^4 d^2 18^{2L}\|\underline{m}\|_1/\epsilon^6), \tag{20}$$

*where* $\tilde{\Theta}$ *hides a logarithmic factor* $\log(d\|\underline{m}\|_1 S \cdot 18^L/(\eta\epsilon))$*. Similar to Theorem 1, the dependency on any of the parameters* $(S, d, 18^L, K, \|\underline{m}\|_1, \epsilon)$ *could potentially be refined by imposing additional assumptions on the* $\psi$ *functions or through a more detailed analysis of approximation and generalization. This question of improving rates remains open for future work.*

*Consider a simplified scenario where* $S = 2$*,* $K$ *is constant, and* $m^{(\ell)} = m \geq 3$ *for each* $\ell \in [L]$*, leading to* $d = m^L$*. In this setup, the sample complexity gives*

$$n = \tilde{\Theta}(L^2 m \cdot d^{2+2\log_m 18}/\epsilon^6) \leq \tilde{\Theta}(L^2 m \cdot d^8/\epsilon^6), \tag{21}$$

*exhibiting a polynomial dependence on* $d$ *and* $1/\epsilon$*. Although the degree of the polynomial is substantial and potentially improvable, this gives the first polynomial sample complexity result for learning the Bayes denoiser in a hierarchical model using the U-Nets.*

**Connection to the task of diffusion generative modeling.** The denoising task examined in this section is closely related to the diffusion model approach (Sohl-Dickstein et al., 2015; Ho et al., 2020; Song & Ermon, 2019; Song et al., 2020) to generative modeling. Diffusion models involve learning a generative model from a dataset of $n$ independent and identically distributed samples $\{\boldsymbol{x}_i\}_{i=1}^n$, drawn from an unknown data distribution $\mu_\star \in \mathcal{P}([S]^d)$. The goal is to generate new samples $\hat{\boldsymbol{x}} \sim \hat{\mu}$ that match the distribution $\mu_\star$. Most diffusion model formulations involve a series of steps that are closely related to the denoising task described earlier. Here, we illustrate how they work using a variant of diffusion models, the stochastic localization process (Eldan, 2013; El Alaoui et al., 2022; Montanari & Wu, 2023; Celentano, 2022; Montanari, 2023):

- **Step 1.** Fit approximate denoising functions $\hat{\boldsymbol{m}}_t : \mathbb{R}^d \to \mathbb{R}^d$ for $t \in [0, T]$. This is done by minimizing the empirical risk over a class of neural networks $\mathcal{F}$ (i.e., the denoising task discussed in this section):

$$\hat{\boldsymbol{m}}_t \equiv \arg\min_{\mathrm{NN}_t \in \mathcal{F}} \frac{1}{nd} \sum_{i=1}^n \|\boldsymbol{x}_i - \mathrm{NN}_t(t \cdot \boldsymbol{x}_i + \sqrt{t} \cdot \boldsymbol{g}_i)\|_2^2, \quad \boldsymbol{g}_i \sim_{iid} \mathcal{N}(\boldsymbol{0}, \mathbf{I}_d). \tag{ERM}$$

- **Step 2.** Simulate a discretized version of a stochastic differential equation (SDE) starting from zero, whose drift term gives the approximate denoising function:

$$\mathrm{d}\boldsymbol{z}_t = \hat{\boldsymbol{m}}_t(\boldsymbol{z}_t) \cdot \mathrm{d}t + \mathrm{d}\boldsymbol{B}_t, \; t \in [0, T], \quad \boldsymbol{z}_0 = \boldsymbol{0}, \tag{SDE}$$

  and generate an approximate sample $\hat{\boldsymbol{x}} = \boldsymbol{z}_T/T \in \mathbb{R}^d$ at the final time $T$.

Standard analysis shows that by replacing the fitted denoising functions $\hat{\boldsymbol{m}}_t(\boldsymbol{z})$ with the true denoising functions $\boldsymbol{m}_t(\boldsymbol{z})$ in Eq. (SDE) and allowing $T \to \infty$, we can effectively recover the original data distribution $\mu_\star$. Consequently, the quality of samples generated from diffusion models hinges on two critical factors: (1) How well the fitted denoising functions $\hat{\boldsymbol{m}}_t$ (ERM) approximate the true denoising functions $\boldsymbol{m}_t$; (2) How accurately the SDE discretization scheme approximates the continuous process as in (SDE).

Recent work has made substantial progress in addressing these two theoretical questions: controlling the SDE discretization error, assuming a reliable denoising function estimator is available (Chen et al., 2022a; 2023a; Lee et al., 2023; Li et al., 2023; Benton et al., 2023); and controlling the denoising function approximation error through neural networks (Oko et al., 2023; Chen et al., 2023b; Mei & Wu, 2023). However, these works have not explained the benefit of employing U-Net in image diffusion modeling, which is the primary focus of the current work. Indeed, by integrating the sample complexity bounds for learning denoising functions, as established in Theorem 2, with standard SDE discretization error bounds, such as the result established in Benton et al. (2023), it is straightforward to derive an end-to-end error bound for the sampling process of diffusion models in GHMs, similar to the strategy of Oko et al. (2023); Chen et al. (2023b); Mei & Wu (2023)[2].

---

[2]We note that the stochastic localization formulation is equivalent to the DDPM diffusion model, differing only in parametrization (Montanari, 2023). In the DDPM model, U-Nets serve to approximate the score function. The score function is a linear combination of the denoising function with an identity map, as per Tweedie's formula. Consequently, Theorem 2 can be readily adapted to establish a sample complexity bound for learning this score function.

**Proof strategy.** The proof strategy for the denoising task parallels that of the classification task. The squared distance between the learned model and the true Bayes denoiser is decomposed into approximation and generalization error terms. The generalization error is bounded via a standard parameter counting argument. The approximation error is controlled by first introducing the belief propagation and message passing algorithm for computing the Bayes denoiser and then showing that U-Nets can effectively approximate this algorithm. A detailed outline of the proof strategy is provided in Appendix A.2, with the complete proof presented in Appendix H.

## 5 FURTHER RELATED WORK

**Generative hierarchical models.** Hierarchical modeling of data distributions has been proposed in a series of works (Mossel, 2016; Poggio et al., 2017; Malach & Shalev-Shwartz, 2020; Schmidt-Hieber, 2020; Allen-Zhu & Li, 2022; Petrini et al., 2023; Sclocchi et al., 2024; Tomasini & Wyart, 2024; Cagnetta & Wyart, 2024; Garnier-Brun et al., 2024; Kadkhodaie et al., 2023a;b). While the hierarchical models in Poggio et al. (2017); Malach & Shalev-Shwartz (2020); Schmidt-Hieber (2020); Allen-Zhu & Li (2022) remain deterministic, Mossel (2016); Petrini et al. (2023); Sclocchi et al. (2024); Tomasini & Wyart (2024); Cagnetta & Wyart (2024); Garnier-Brun et al. (2024) studied the generative version of hierarchical models. The diffusion model for multi-scale image distribution representations has been empirically examined in Kadkhodaie et al. (2023a;b), which demonstrated that U-Nets are effective in modeling denoising algorithms. The theoretical and empirical evidence presented in Sclocchi et al. (2024); Tomasini & Wyart (2024); Petrini et al. (2023) underscores the effectiveness of generative hierarchical models in capturing the combinatorial properties of image datasets. Given their significant relevance to this work, we delve deeper into these studies.

**Contributions of Petrini et al. (2023); Sclocchi et al. (2024); Tomasini & Wyart (2024).** The series of works on hierarchical generative models (Petrini et al., 2023; Sclocchi et al., 2024; Tomasini & Wyart, 2024) inspired the current study. Sclocchi et al. (2024) first pointed out that the belief propagation denoising algorithm of hierarchical models consists of downward and upward processes. Through mean-field analysis on a random generative hierarchical model, they identified a phase transition phenomenon, aligning with empirical observations in diffusion models, thereby providing strong evidence of the efficacy of these models in handling combinatorial data properties. Petrini et al. (2023), on the other hand, first introduced these models in a classification context. Petrini et al. (2023); Tomasini & Wyart (2024) demonstrated that learning hierarchical models using multi-layer networks circumvents the curse of dimensionality. Specifically, they theoretically and empirically characterized the sample complexity, showing that it remains polynomial in dimension when learning convolutional networks under random generative rules. On the other hand, in the absence of correlations, they showed that the sample complexity is again exponential in the dimension, even for hierarchical generative models. This learning incapability is not captured by our analysis which does not consider optimization.

**ConvNets and U-Nets and their implicit bias.** Convolutional networks (LeCun et al., 1989; 1998; Krizhevsky et al., 2012; Szegedy et al., 2015; He et al., 2016) have become the state-of-the-art architecture for image classification and have been the backbone for many computer vision tasks. U-Nets (Ronneberger et al., 2015; Zhou et al., 2018; Siddique et al., 2021; Oktay et al., 2018) have been particularly well-suited for image segmentation and denoising tasks (Ronneberger et al., 2015), and have served as the backbone architecture for diffusion models (Sohl-Dickstein et al., 2015; Ho et al., 2020; Song & Ermon, 2019; Song et al., 2020). A series of theoretical works has explained the inductive bias of CNNs (Bruna & Mallat, 2013; Gunasekar et al., 2018; Bietti & Mairal, 2019b;a; Scetbon & Harchaoui, 2020; Li et al., 2020; Bietti, 2021; Mei et al., 2021; Misiakiewicz & Mei, 2022; Cagnetta et al., 2023; Favero et al., 2021; Bietti et al., 2021; Xiao, 2022; Petrini et al., 2023; Tomasini & Wyart, 2024; Wang & Wu, 2024). However, they mostly focused on the classification and regression setting and were not concerned with the role of U-Nets in denoising tasks. The implicit bias of U-Nets has been theoretically investigated in Williams et al. (2024); Falck et al. (2022), where they found that the U-Nets are conjugate to the ResNets. In contrast, we demonstrate that U-Nets can effectively approximate the belief propagation denoising algorithms of GHMs. We note that Cui et al. (2023) analyzed the learning dynamics for a simple U-Net in diffusion models.

**Neural networks approximation of algorithms.** A recent line of work has investigated the expressiveness of neural networks through an algorithm approximation viewpoint (Wei et al., 2022; Bai et al., 2024; Giannou et al., 2023; Liu et al., 2022a; Marwah et al., 2021; 2023; Lin et al., 2023; Mei & Wu, 2023). In particular, Wei et al. (2022); Bai et al. (2024); Giannou et al. (2023); Liu et al. (2022a); Lin et al. (2023) demonstrate that transformers can efficiently approximate several algorithm classes, such as gradient descent, reinforcement learning algorithms, and even Turing machines. In the context of diffusion models, Mei & Wu (2023) shows that ResNets can efficiently approximate the score function of high-dimensional graphical models by approximating the variational inference algorithm. Our work is closely related to Mei & Wu (2023), except that we study neural network approximation in a different statistical model and network architecture.

From a practical viewpoint, a line of work has focused on neural network denoising by unrolling iterative denoising algorithms into deep networks (Gregor & LeCun, 2010; Zheng et al., 2015; Zhang & Ghanem, 2018; Papyan et al., 2017; Ma et al., 2021; Chen et al., 2018; Borgerding et al., 2017; Monga et al., 2021; Yu et al., 2023a;b). While this literature has primarily focused on devising better denoising algorithms, our work leverages this perspective to develop neural network approximation theory and explain existing network architectures.

**Related theory of diffusion models.** In recent years, diffusion models (Sohl-Dickstein et al., 2015; Ho et al., 2020; Song & Ermon, 2019; Song et al., 2020) have emerged as a leading approach for generative modeling. Neural network-based score function approximation has been recently studied from the function approximation viewpoint in Oko et al. (2023); Chen et al. (2023b); Yuan et al. (2023); Shah et al. (2023); Biroli & Mézard (2023), and from the algorithm approximation viewpoint in Mei & Wu (2023). Theoretical studies of other aspects of diffusion models include Liu et al. (2022b); Li et al. (2023); Lee et al. (2023); Chen et al. (2022b; 2023d; 2022a; 2023c;a); Benton et al. (2023); El Alaoui et al. (2022); Montanari & Wu (2023); Celentano (2022); Ghio et al. (2023); Biroli & Mézard (2023); Biroli et al. (2024); Cui et al. (2023); Fu et al. (2024); Wu et al. (2024). For a comprehensive introduction to the theory of diffusion models, see the recent review (Chen et al., 2024).

## 6 CONCLUSIONS AND DISCUSSIONS

In this paper, we introduced a novel interpretation of the U-Net architecture through the lens of generative hierarchical models. We demonstrated that their belief propagation denoising algorithm naturally induces the encoder-decoder structure, the long skip connections, and the pooling and upsampling operations of the U-Nets. We also provided an efficient sample complexity bound for learning the denoising function with U-Nets. Furthermore, we discussed the broader implications of these findings for diffusion models. We also showed that ConvNets are well-suited for classification tasks within these models. Our study offers a unified perspective on the roles of ConvNets and U-Nets, highlighting the versatility of generative hierarchical models in capturing complex data distributions across language and image domains.

The results presented in this paper offer considerable scope for enhancement. We initially assumed that the covariates $x$ lie in the discrete space $[S]^d$, and extending these results to continuous spaces would be an intriguing direction for future research. Additionally, the dependencies of the sample complexity bound on $d$ and $1/\epsilon$ may be amenable to improvement through more careful analysis. Moreover, the convolution operations employed in this paper are different from those commonly employed in practical settings. It would be worthwhile to explore graphical models where the belief propagation algorithm aligns more naturally with ConvNets and U-Nets that utilize standard convolution operations.

On the practical side, our theoretical findings generated a hypothesis of the functionality of each layer of the U-Nets. Verifying these hypotheses in pre-trained U-Nets, such as those used in stable diffusion models, using interpretability methods, could yield valuable insights. Furthermore, extending these results to include conditional denoising functions represents an exciting direction for future research. Finally, we hope that the insights provided in this paper could guide the design of innovative network architectures.

ACKNOWLEDGEMENT

This work is supported by NSF DMS-2210827, CCF-2315725, CAREER DMS-2339904, ONR N00014-24-S-B001, a Sloan Research Fellow Award, an Amazon Research Award, a Google Research Scholar Award, and an Okawa Foundation Research Grant. The author would like to thank Hui Xu, Yu Bai, and Yuchen Wu for their valuable discussions.

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

CONTENTS

## A    PROOF STRATEGY

### A.1    PROOF STRATEGY: CONVNETS APPROXIMATE THE BELIEF PROPAGATION ALGORITHM

Lemma 17 introduces a decomposition of $D_2^2(\mu_{\mathrm{NN}}^{\widehat{W}}, \mu_\star)$ into two components, approximation error and generalization error:

$$D_2^2(\mu_{\mathrm{NN}}^{\widehat{W}}, \mu_\star) \leq \underbrace{\inf_{W \in \mathcal{W}} D_2^2(\mu_{\mathrm{NN}}^{W}, \mu_\star)}_{\text{approximation error}} + 2 \cdot \underbrace{\sup_{W \in \mathcal{W}} \left| \widehat{R}(\mu_{\mathrm{NN}}^{W}) - R(\mu_{\mathrm{NN}}^{W}) \right|}_{\text{generalization error}}.$$

The bound of generalization error follows a standard approach: employing a chaining argument, the error is controlled by $\tilde{O}(\sqrt{d_{\mathrm{p}}/n})$, where $d_{\mathrm{p}}$ represents the number of parameters in the ConvNet class.

In the following, we describe our strategy to control the approximation error: we first present the belief propagation and message passing algorithm for computing the Bayes classifier $\mu_\star$, and then demonstrate that ConvNets are capable of effectively approximating this message passing algorithm.

**The belief propagation and message passing algorithm.**    The belief propagation algorithm operates on input $x \in [S]^d$ and iteratively calculates the beliefs $\{\nu_v^{(\ell)} \in \Delta([S])\}_{\ell \in \{0,\dots,L\}, v \in \mathcal{V}^{(\ell)}}$ as follows:

$$\nu_v^{(L)}(x_v^{(L)}) = 1\{x_v^{(L)} = x_v\},$$
$$\nu_v^{(\ell)}(x_v^{(\ell)}) \propto \sum_{x_{\mathcal{C}(v)}^{(\ell+1)}} \prod_{v' \in \mathcal{C}(v)} \left( \psi_{\iota(v')}^{(\ell+1)}(x_v^{(\ell)}, x_{v'}^{(\ell+1)}) \nu_{v'}^{(\ell+1)}(x_{v'}^{(\ell+1)}) \right), \quad \ell = L-1, \dots, 0,$$
$$\mu_{\mathrm{BP}}(y|x) = \nu_{\mathrm{r}}^{(0)}(y).$$

(BP-CLS)

Classical results in graphical models verify that the belief propagation algorithm accurately computes the Bayes classifier in this tree graph.

**Lemma 1** (BP calculates the Bayes classifier exactly (Pearl, 1982; Wainwright et al., 2008; Mezard & Montanari, 2009))**.** *When applying the belief propagation algorithm (BP-CLS) starting with $x \in [S]^d$, it holds that $\mu_\star(\cdot|x) = \mu_{\mathrm{BP}}(\cdot|x)$.*

The belief propagation algorithm can be streamlined into a message passing algorithm, starting with the initialization $h_v^{(L)} = x_v$ for each node $v$ in the highest layer $\mathcal{V}^{(L)}$. The operations are defined as follows:

$$q_v^{(\ell)} = f_{\iota(v)}^{(\ell)}(h_v^{(\ell)}) \in \mathbb{R}^S, \qquad\qquad \ell \in [L], \quad v \in \mathcal{V}^{(\ell)},$$
$$h_v^{(\ell-1)} = \mathrm{normalize}\left( \sum_{v' \in \mathcal{C}(v)} q_{v'}^{(\ell)} \right) \in \mathbb{R}^S, \qquad \ell \in [L], \quad v \in \mathcal{V}^{(\ell-1)}, \qquad \text{(MP-CLS)}$$
$$\mu_{\mathrm{MP}}(y|x) = \mathrm{softmax}(h_{\mathrm{r}}^{(0)}).$$

The functions $f_\iota^{(L)} : [S] \to \mathbb{R}^S$, $\{f_\iota^{(\ell)} : \mathbb{R}^S \to \mathbb{R}^S\}_{\ell \in [L-1]}$ are defined as:

$$\begin{aligned} f_\iota^{(L)}(x)_s &= \log \psi_\iota^{(L)}(s, x), & x \in [S], s \in [S], \\ f_\iota^{(\ell)}(h)_s &= \log \sum_{a \in [S]} \psi_\iota^{(\ell)}(s, a) e^{h_a}, & h \in \mathbb{R}^S, s \in [S], \ell \in [L-1]. \end{aligned}$$

(22)

We note that the normalization operator in (MP-CLS) is non-essential and could be dropped; however, we include it to ensure the formulation closely mirrors (ConvNet), offering a technical benefit.

The subsequent proposition affirms that message passing is essentially equivalent to belief propagation:

**Proposition 3** (BP reduces to MP)**.** *Consider the belief propagation algorithm and the message passing algorithm, both starting from $x \in [S]^d$, as in Eq. (BP-CLS) and (MP-CLS). Then we have $\nu_v^{(\ell)}(\cdot) = \mathrm{softmax}(h_v^{(\ell)})$ for all $0 \leq \ell \leq L-1$ and $v \in \mathcal{V}^{(\ell)}$. In particular, we have $\mu_{\mathrm{BP}}(\cdot|x) = \mu_{\mathrm{MP}}(\cdot|x) = \mu_\star(\cdot|x)$.*

The proof of Proposition 3 is presented in Section C.

**Approximating message passing with ConvNets.** By comparing the message passing algorithm (MP-CLS) alongside the ConvNet (ConvNet), the primary distinction lies in the nonlinear functions used: $f_\iota^{(\ell)}$ versus $\mathrm{NN}_\iota^{(\ell)}$. Given the expression $f_\iota^{(\ell)}(h)_s = \log \sum_{a \in [S]} \psi_\iota^{(\ell)}(s, a) e^{h_a}$, it becomes evident that approximating the logarithmic and exponential functions using one-hidden-layer networks enables $f_\iota^{(\ell)}(h)$ to be effectively approximated by a two-hidden-layer neural network. This leads to the following theorem:

**Theorem 4** (ConvNets approximation of Bayes classifier). *Let Assumption 1 and 2 hold. For any $\delta > 0$, take*

$$D = 4\lceil S^2 K^2 d \cdot 3^L / \delta \rceil, \quad B = \mathrm{Poly}(d, S, K, 3^L, 1/\delta).$$

*Then there exists $\boldsymbol{W} \in \mathcal{W}_{d,\underline{m},L,S,D,B}$ as in Eq. (9), such that defining $\mu_{\mathrm{NN}}^{\boldsymbol{W}}$ as in Eq. (ConvNet), we have*

$$\max_{y \in [S], \boldsymbol{x} \in [S]^d} \left| \log \mu_\star(y|\boldsymbol{x}) - \log \mu_{\mathrm{NN}}^{\boldsymbol{W}}(y|\boldsymbol{x}) \right| \le \delta.$$

The proof of Theorem 4 is detailed in Section D.

### A.2 PROOF STRATEGY: U-NETS APPROXIMATE THE BELIEF PROPAGATION ALGORITHM

The proof strategy for the denoising task closely aligns with that of the classification task as detailed in Section A.1.

Lemma 22 introduces a decomposition of $\mathrm{D}_2^2(\boldsymbol{m}_{\mathrm{NN}}^{\widehat{\boldsymbol{W}}}, \boldsymbol{m}_\star)$ into two components, approximation error and generalization error:

$$\mathrm{D}_2^2(\boldsymbol{m}_{\mathrm{NN}}^{\widehat{\boldsymbol{W}}}, \boldsymbol{m}_\star) \le \underbrace{\inf_{\boldsymbol{W} \in \mathcal{W}} \mathrm{D}_2^2(\boldsymbol{m}_{\mathrm{NN}}^{\boldsymbol{W}}, \boldsymbol{m}_\star)}_{\text{approximation error}} + 2 \cdot \underbrace{\sup_{\boldsymbol{W} \in \mathcal{W}} \left| \widehat{\mathrm{R}}(\boldsymbol{m}_{\mathrm{NN}}^{\boldsymbol{W}}) - \mathrm{R}(\boldsymbol{m}_{\mathrm{NN}}^{\boldsymbol{W}}) \right|}_{\text{generalization error}}.$$

The bound of generalization error follows a standard approach: employing a chaining argument, the error is controlled by $\tilde{O}(\sqrt{d_{\mathrm{p}}/n})$, where $d_{\mathrm{p}}$ represents the number of parameters in the U-Net class.

In the following, we describe our strategy to control the approximation error: we first present the belief propagation and message passing algorithm for computing the Bayes denoiser $\boldsymbol{m}_\star$, and then demonstrate that U-Nets are capable of effectively approximating this message passing algorithm.

**The belief propagation and message passing algorithm.** The belief propagation algorithm operates on input $\boldsymbol{z} \in \mathbb{R}^d$ and iteratively calculates the beliefs $\{\nu_{\downarrow,v}^{(\ell)}, \nu_{\uparrow,v}^{(\ell)} \in \Delta([S])\}_{\ell \in \{0,\dots,L\}, v \in \mathcal{V}^{(\ell)}}$ as follows:

$$\nu_{\downarrow,v}^{(L)}(x_v^{(L)}) = \psi^{(L+1)}(x_v^{(L)}, z_v),$$

$$\nu_{\downarrow,v}^{(\ell)}(x_v^{(\ell)}) \propto \sum_{x_{\mathcal{C}(v)}^{(\ell+1)}} \prod_{v' \in \mathcal{C}(v)} \left( \psi_{\iota(v')}^{(\ell+1)}(x_v^{(\ell)}, x_{v'}^{(\ell+1)}) \nu_{\downarrow,v'}^{(\ell+1)}(x_{v'}^{(\ell+1)}) \right), \qquad \ell = L-1, \dots, 0,$$

$$\nu_{\uparrow,\mathrm{r}}^{(0)}(x_\mathrm{r}^{(0)}) \propto 1,$$

$$\nu_{\uparrow,v}^{(\ell)}(x_v^{(\ell)}) \propto \sum_{x_{\mathrm{pa}(v)}^{(\ell-1)}, x_{\mathcal{N}(v)}^{(\ell)}} \psi^{(\ell)}(x_{\mathrm{pa}(v)}^{(\ell-1)}, x_{\mathcal{C}(\mathrm{pa}(v))}^{(\ell)}) \nu_{\uparrow,\mathrm{pa}(v)}^{(\ell-1)}(x_{\mathrm{pa}(v)}^{(\ell-1)}) \prod_{v' \in \mathcal{N}(v)} \nu_{\downarrow,v'}^{(\ell)}(x_{v'}^{(\ell)}), \quad \ell = 1, \dots, L,$$

$$\nu_v^{(L)}(x_v^{(L)}) \propto \nu_{\uparrow,L}(x_v^{(L)}) \psi^{(L+1)}(x_v^{(L)}, z_v),$$

$$\boldsymbol{m}_{\mathrm{BP}}(\boldsymbol{z})_v = \sum_{x_v^{(L)}} x_v^{(L)} \nu_v^{(L)}(x_v^{(L)}), \qquad\qquad\qquad\qquad\qquad\qquad \text{(BP-DNS)}$$

where $\psi^{(L+1)}(x_v^{(L)}, z_v) := \exp\{-(x_v^{(L)} - z_v)^2/2\}$. Classical results in graphical models verify that the belief propagation algorithm accurately computes the Bayes denoiser in this tree graph.

**Lemma 2** (BP calculates the Bayes denoiser exactly (Pearl, 1982; Wainwright et al., 2008; Mezard & Montanari, 2009)). *When applying the belief propagation algorithm (BP-DNS) starting with $\boldsymbol{z} \in \mathbb{R}^d$, it holds that $\mu_\star(x_v|\boldsymbol{z}) = \nu_v^{(L)}(x_v)$ for $v \in \mathcal{V}^{(L)}$, so that $\boldsymbol{m}_{\mathrm{BP}}(\boldsymbol{z}) = \boldsymbol{m}(\boldsymbol{z})$.*

We remark that the downward-upward structure of belief propagation in generative hierarchical models has been pointed out in the literature (Sclocchi et al., 2024).

The belief propagation algorithm can be streamlined into a message passing algorithm, starting with the initialization $h_{\downarrow,v}^{(L)} = (-(x-z_v)^2/2)_{x\in[S]} \in \mathbb{R}^S$ for each node $v$ in the highest layer $\mathcal{V}^{(L)}$. The operations are defined as follows:

$$
\begin{aligned}
q_{\downarrow,v}^{(\ell)} &= f_{\downarrow,\iota(v)}^{(\ell)}(\text{normalize}(h_{\downarrow,v}^{(\ell)})) \in \mathbb{R}^S, & \ell \in [L], \quad v \in \mathcal{V}^{(\ell)}, \\
h_{\downarrow,v}^{(\ell-1)} &= \sum_{v'\in\mathcal{C}(v)} q_{\downarrow,v'}^{(\ell)} \in \mathbb{R}^S, & \ell \in [L], \quad v \in \mathcal{V}^{(\ell-1)}, \\
u_{\uparrow,v}^{(\ell)} &= b_{\uparrow,\text{pa}(v)}^{(\ell-1)} \in \mathbb{R}^S, \quad (\text{with } b_{\uparrow,\text{r}}^{(0)} = h_{\downarrow,\text{r}}^{(0)}) & \ell \in [L], \quad v \in \mathcal{V}^{(\ell)}, \quad \text{(MP-DNS)} \\
b_{\uparrow,v}^{(\ell)} &= f_{\uparrow,\iota(v)}^{(\ell)}(\text{normalize}(u_{\uparrow,v}^{(\ell)} - q_{\downarrow,v}^{(\ell)})) + h_{\downarrow,v}^{(\ell)} \in \mathbb{R}^S, & \ell \in [L], \quad v \in \mathcal{V}^{(\ell)}, \\
\boldsymbol{m}_{\text{MP}}(\boldsymbol{z})_v &= \sum_{s\in[S]} s \cdot \text{softmax}(b_{\uparrow,v}^{(L)})_s, & v \in \mathcal{V}^{(L)}.
\end{aligned}
$$

The functions $\{f_{\downarrow,\iota}^{(\ell)}, f_{\uparrow,\iota}^{(\ell)} : \mathbb{R}^S \to \mathbb{R}^S\}_{\ell\in[L]}$ are defined as

$$
\begin{aligned}
f_{\downarrow,\iota}^{(\ell)}(h)_s &= \log \sum_{a\in[S]} \psi_\iota^{(\ell)}(s,a) e^{h_a}, & h \in \mathbb{R}^S, s \in [S], \ell \in [L], \\
f_{\uparrow,\iota}^{(\ell)}(h)_s &= \log \sum_{a\in[S]} \psi_\iota^{(\ell)}(a,s) e^{h_a}, & h \in \mathbb{R}^S, s \in [S], \ell \in [L].
\end{aligned} \tag{23}
$$

We note that the normalization operator in (MP-DNS) is non-essential and could be dropped; however, we include it to ensure the formulation closely mirrors (UNet), offering a technical benefit.

The subsequent proposition affirms that message passing is essentially equivalent to belief propagation:

**Proposition 5** (BP reduces to MP). *Consider the belief propagation algorithm and the message passing algorithm, both with input $\boldsymbol{z} \in \mathbb{R}^d$, as in Eq. (BP-DNS) and (MP-DNS). Then we have $\nu_{\downarrow,v}^{(\ell)}(\cdot) = \text{softmax}(h_{\downarrow,v}^{(\ell)})$ and $\nu_{\uparrow,v}^{(\ell)}(\cdot) = \text{softmax}(b_{\uparrow,v}^{(\ell)} - h_{\downarrow,v}^{(\ell)})$, and $\nu_v^{(L)}(\cdot) = \text{softmax}(b_{\uparrow,v}^{(L)})$. In particular, $\boldsymbol{m}_{\text{MP}}(\boldsymbol{z}) = \boldsymbol{m}_{\text{BP}}(\boldsymbol{z}) = \boldsymbol{m}(\boldsymbol{z})$.*

The proof of Proposition 5 is presented in Section F.

**Approximating message passing with ConvNets.** By comparing the message passing algorithm (MP-DNS) alongside the U-Net (UNet), the primary distinction lies in the nonlinear functions used: $f_{\diamond,\iota}^{(\ell)}$ versus $\text{NN}_{\diamond,\iota}^{(\ell)}$. Notably, $f_{\diamond,\iota}^{(\ell)}$ entails a log-sum-exponential structure. This structure suggests that approximating the logarithmic and exponential functions with one-hidden-layer neural networks can allow $f_{\diamond,\iota}^{(\ell)}(h)$ to be effectively approximated by a two-hidden-layer neural network. This leads to the following theorem:

**Theorem 6** (U-Nets approximation of Bayes denoiser). *Let Assumption 1 and 2 hold. For any $\delta > 0$, take*

$$
D = 4\lceil S^3 K^2 d \cdot 18^L/\delta \rceil, \quad B = \text{Poly}(d, S, K, 18^L, 1/\delta).
$$

*Then there exists $\boldsymbol{W} \in \mathcal{W}_{d,\underline{m},L,S,D,B}$ as in Eq. (18), such that defining $\boldsymbol{m}_{\text{NN}}^{\boldsymbol{W}}$ as in Eq. (UNet), we have*

$$
\sup_{\boldsymbol{z}\in\mathbb{R}^d} \|\boldsymbol{m}(\boldsymbol{z}) - \boldsymbol{m}_{\text{NN}}(\boldsymbol{z})\|_\infty \le \delta.
$$

The proof of Theorem 6 is detailed in Section G.

## B  TECHNICAL PRELIMINARIES

We here present a bound on the supremum of sub-Gaussian processes, whose proof was based on the chaining argument.

**Lemma 3** (Proposition A.4 of (Bai et al., 2024)). *Suppose that $\{X_w\}_{w\in\Theta}$ is a zero-mean random process given by*

$$
X_w \equiv \frac{1}{n}\sum_{i=1}^n f(z_i;w) - \mathbb{E}_z[f(z;w)],
$$

*where $z_1,\cdots,z_n$ are i.i.d samples from a distribution $\mathbb{P}_z$ such that the following assumption holds:*

(a) *The index set $\Theta$ is equipped with a distance $\rho$ and diameter $B_{\mathrm{p}}$. Further, assume that for some constant $A_{\mathrm{p}}$, for any ball $\Theta'$ of radius $r$ in $\Theta$, the covering number admits upper bound $\log N(\Delta; \Theta', \rho) \le d_{\mathrm{p}} \log(2A_{\mathrm{p}}r/\Delta)$ for all $0 < \Delta \le 2r$.*

(b) *For any fixed $w \in \Theta$ and $z$ sampled from $\mathbb{P}_z$, the random variable $f(z; w) - \mathbb{E}_z[f(z; w)]$ is a $\sigma$-sub-Gaussian random variable ($\mathbb{E}[e^{\lambda[f(z;w) - \mathbb{E}_{z'}[f(z';w)]]}] \le e^{\lambda^2 \sigma^2/2}$ for any $\lambda \in \mathbb{R}$).*

(c) *For any $w, w' \in \Theta$ and $z$ sampled from $\mathbb{P}_z$, the random variable $f(z; w) - f(z; w')$ is a $\sigma'\rho(w, w')$-sub-Gaussian random variable ($\mathbb{E}[e^{\lambda[f(z;w) - f(z;w')]}] \le e^{\lambda^2 (\sigma')^2 \rho^2(w,w')/2}$ for any $\lambda \in \mathbb{R}$).*

*Then with probability at least $1 - \eta$, it holds that*

$$\sup_{w \in \Theta} |X_w| \le C\sigma \sqrt{\frac{d_{\mathrm{p}} \cdot \log(2A_{\mathrm{p}}(1 + B_{\mathrm{p}}\sigma'/\sigma)) + \log(1/\eta)}{n}},$$

*where $C$ is a universal constant.*

We next present a simple inequality used in the proof of Theorem 1.

**Lemma 4** (From log ratio bound to square distance bound). *Let $p$ and $q$ be two probability measures on $\Delta([S])$ such that*

$$\max_{y \in [S]} \left| \log p(y) - \log q(y) \right| \le \delta.$$

*Then we have*

$$\sum_{s=1}^{S} (p(s) - q(s))^2 \le (e^\delta - 1)^2.$$

*Proof of Lemma 4.* The lemma is by the fact that $|p(y) - q(y)| \le (\exp\{|\log p(y) - \log q(y)|\} - 1) \cdot p(y)$. $\square$

## C  PROOF OF PROPOSITION 3

*Proof of Proposition 3.* By Eq. (MP-CLS) and (22), defining $\nu_v^{(\ell)}(\cdot) = \mathrm{softmax}(h_v^{(\ell)})$, then for $\ell \le L - 2$, we get

$$\nu_v^{(\ell)}(x_v^{(\ell)}) \propto \prod_{v' \in \mathcal{C}(v)} \left( \sum_{a \in [S]} \psi_{\iota(v')}^{(\ell+1)}(x_v^{(\ell)}, a) e^{(h_v^{(\ell+1)})_a} \right)$$

$$\propto \sum_{x_{\mathcal{C}(v)}^{(\ell+1)}} \prod_{v' \in \mathcal{C}(v)} \left( \psi_{\iota(v')}^{(\ell+1)}(x_v^{(\ell)}, x_{v'}^{(\ell+1)}) \nu_{v'}^{(\ell+1)}(x_{v'}^{(\ell+1)}) \right).$$

This coincides with the update rule in Eq. (BP-CLS). For $\ell = L - 1$, we get

$$\nu_v^{(L-1)}(x_v^{(L-1)}) \propto \prod_{v' \in \mathcal{C}(v)} \left( \sum_{a \in [S]} \psi_{\iota(v')}^{(L)}(x_v^{(L-1)}, a) 1\{a = x_{v'}\} \right) \propto \prod_{v' \in \mathcal{C}(v)} \psi_{\iota(v')}^{(L)}(x_v^{(L-1)}, x_{v'}).$$

This again coincides with the update rule in Eq. (BP-CLS). This finishes the proof of Proposition 3. $\square$

## D  PROOF OF THEOREM 4

*Proof of Theorem 4.* By Lemma 13, take $M_1^{(\ell)} = \lceil 2SKd3^L/\delta \rceil + 1$ and $M_2^{(\ell)} = \lceil 2SK^2d3^L/\delta \rceil + 1$. Then there exists $\{(a_j, w_j, b_j)\}_{j \in [M_1^{(\ell)}]}$ and $\{(\bar{a}_j, \bar{w}_j, \bar{b}_j)\}_{j \in [M_2^{(\ell)}]}$ with

$$\sup_j |a_j| \le 2SK, \quad \sup_j |w_j| \le 1, \quad \sup_j |b_j| \le SK, \quad \sup_j |\bar{a}_j| \le 2, \quad \sup_j |\bar{w}_j| \le 1, \quad \sup_j |\bar{b}_j| \le \log(4 \cdot 3^L SdK^2/\delta),$$

such that defining

$$\log_{\delta \star}(x) = \sum_{j=1}^{M_1^{(\ell)}} a_j \cdot \mathrm{ReLU}(w_j x + b_j), \quad \exp_{\delta \star}(x) = \sum_{j=1}^{M_2^{(\ell)}} \bar{a}_j \cdot \mathrm{ReLU}(\bar{w}_j x + \bar{b}_j),$$

and defining $f_\iota^{(\ell)}(h), \bar{f}_\iota^{(\ell)}(h) \in \mathbb{R}^S$ by

$$f_\iota^{(\ell)}(h)_i = \log \sum_{j=1}^S \psi_\iota^{(\ell)}(i,j) \exp(h_j), \quad \bar{f}_\iota^{(\ell)}(h)_i = \log_{\delta_\star} \sum_{j=1}^S \psi_\iota^{(\ell)}(i,j) \exp_{\delta_\star}(h_j), \quad \forall i \in [S],$$

we have

$$\sup_{\max_i h_i = 0} \|f_\iota^{(\ell)}(h) - \bar{f}_\iota^{(\ell)}(h)\|_\infty \le \delta/(d3^L). \tag{24}$$

In addition, by Lemma 15, take $M_1^{(L)} = \lceil 3^L dK/\delta \rceil + 1$ and $M_2^{(L)} = 3$. Then there exists $\{(a_j^{(L)}, w_j^{(L)}, b_j^{(L)})\}_{j \in [M_1^{(L)}]}$ and $\{(\bar{a}_j^{(L)}, \bar{w}_j^{(L)}, \bar{b}_j^{(L)})\}_{j \in [M_2^{(L)}]}$ with

$$\sup_j |a_j^{(L)}| \le 2K, \quad \sup_j |w_j^{(L)}| \le 1, \quad \sup_j |b_j^{(L)}| \le SK, \quad \sup_j |\bar{a}_j^{(L)}| \le 4, \quad \sup_j |\bar{w}_j^{(L)}| \le 1, \quad \sup_j |\bar{b}_j^{(L)}| \le 1,$$

such that defining

$$\log_\delta(x) = \sum_{j=1}^{M_1^{(L)}} a_j^{(L)} \cdot \mathrm{ReLU}(w_j^{(L)} x + b_j^{(L)}), \quad \mathrm{Ind}(x) = \sum_{j=1}^{M_2^{(L)}} \bar{a}_j^{(L)} \cdot \mathrm{ReLU}(\bar{w}_j^{(L)} x + \bar{b}_j^{(L)}),$$

and defining $f_\iota^{(L)}(h), \bar{f}_\iota^{(L)}(h) \in \mathbb{R}^S$ by

$$f_\iota^{(L)}(x)_i = \log \sum_{j=1}^S \psi_\iota^{(L)}(i,j) 1(x=j), \quad \bar{f}_\iota^{(L)}(x)_i = \log_\delta \sum_{j=1}^S \psi_\iota^{(L)}(i,j) \mathrm{Ind}(x-j), \quad \forall i \in [S],$$

we have

$$\sup_{x \in [S]} \|f_\iota^{(L)}(x) - \bar{f}_\iota^{(L)}(x)\|_\infty \le \delta/(d3^L). \tag{25}$$

By Eq. (24) and (25) and Lemma 5, taking $h_r^{(0)} \in \mathbb{R}^S$ to be as defined in Eq. (MP-CLS) and $\bar{h}_r^{(0)} \in \mathbb{R}^S$ to be as defined in Eq. (A-MP-CLS) with $\{\bar{f}_\iota^{(\ell)}\}_{\ell \in [L], \iota \in [m^{(\ell)}]}$ as defined above, we have

$$\|h_r^{(0)} - \bar{h}_r^{(0)}\|_\infty \le [\delta/(d3^L)] \times \prod_{1 \le \ell \le L} (2m^{(\ell)} + 1) \le \delta.$$

As a consequence, we just need to show that the approximate version of message passing algorithm as in Eq. (A-MP-CLS) could be cast as a neural network.

Indeed, by Lemma 14, there exist two-hidden-layer neural networks (for $\ell \in [L-1]$ and $\iota \in [m^{(\ell)}]$)

$$\mathrm{NN}_{W_{1,\iota}^{(\ell)}, W_{2,\iota}^{(\ell)}, W_{3,\iota}^{(\ell)}}(h) = W_{1,\iota}^{(\ell)} \cdot \mathrm{ReLU}(W_{2,\iota}^{(\ell)} \cdot \mathrm{ReLU}(W_{3,\iota}^{(\ell)} \cdot [h;1])),$$

with $W_{1,\iota}^{(\ell)} \in \mathbb{R}^{S \times SM_1^{(\ell)}}, W_{2,\iota}^{(\ell)} \in \mathbb{R}^{SM_1^{(\ell)} \times (SM_2^{(\ell)}+1)}, W_{3,\iota}^{(\ell)} \in \mathbb{R}^{(SM_2^{(\ell)}+1) \times (S+1)}$, and

$$\|W_{1,\iota}^{(\ell)}\|_{\max} \le 2SK, \quad \|W_{2,\iota}^{(\ell)}\|_{\max} \le \mathrm{Poly}(SKM_1^{(\ell)} M_2^{(\ell)}), \quad \|W_{3,\iota}^{(\ell)}\|_{\max} \le \log(4SK^2/\delta).$$

such that

$$\mathrm{NN}_{W_{1,\iota}^{(\ell)}, W_{2,\iota}^{(\ell)}, W_{3,\iota}^{(\ell)}}(h) = \bar{f}_\iota^{(\ell)}(h), \quad \forall h \in \mathbb{R}^S \text{ such that } \max_j h_j = 0.$$

Furthermore, by Lemma 16, there exist two-hidden-layer neural networks (for $\iota \in [m^{(L)}]$)

$$\mathrm{NN}_{W_{1,\iota}^{(L)}, W_{2,\iota}^{(L)}, W_{3,\iota}^{(L)}}(x) = W_{1,\iota}^{(L)} \cdot \mathrm{ReLU}(W_{2,\iota}^{(L)} \cdot \mathrm{ReLU}(W_{3,\iota}^{(L)} \cdot [x;1])),$$

with $W_{1,\iota}^{(L)} \in \mathbb{R}^{S \times SM_1^{(L)}}, W_{2,\iota}^{(L)} \in \mathbb{R}^{SM_1^{(L)} \times (SM_2^{(L)}+1)}, W_{3,\iota}^{(L)} \in \mathbb{R}^{(SM_2^{(L)}+1) \times 2}$,

$$\|W_{1,\iota}^{(L)}\|_{\max} \le 2K, \quad \|W_{2,\iota}^{(L)}\|_{\max} \le \mathrm{Poly}(SKM_1^{(L)} M_2^{(L)}), \quad \|W_{3,\iota}^{(L)}\|_{\max} \le 1.$$

such that

$$\mathrm{NN}_{W_{1,\iota}^{(L)}, W_{2,\iota}^{(L)}, W_{3,\iota}^{(L)}}(x) = \bar{f}_\iota^{(L)}(x), \quad \forall x \in [S].$$

This proves that the approximate version of message passing as in Eq. (A-MP-CLS) can be cast into the convolutional neural network as in Eq. (ConvNet) with proper choice of dimension

$$D \ge \max_{\ell \in [L]} \{SM_1^{(\ell)}, SM_2^{(\ell)} + 1\} = S \times \left(\lceil 2SK^2 d3^L/\delta \rceil + 1\right) + 1,$$

and norm of the weights. This finishes the proof of Theorem 4. $\qquad\square$

### D.1 AUXILLARY LEMMAS

**Lemma 5** (Error propagation of the approximate version of message passing in classification). *Assume we have functions $f_\iota^{(\ell)}$ and $\bar{f}_\iota^{(\ell)}$ such that*

$$\|f_\iota^{(L)}(x) - \bar{f}_\iota^{(L)}(x)\|_\infty \leq \delta, \quad \forall x \in [S],$$
$$\|f_\iota^{(\ell)}(h) - \bar{f}_\iota^{(\ell)}(h)\|_\infty \leq \delta, \quad \forall h \in \mathbb{R}^S \text{ such that } \max_{j \in [S]} h_j = 0, \ \forall \ell \leq L - 1. \tag{26}$$

*Furthermore, consider the following approximate version of message passing algorithm with initialization $\bar{h}_v^{(L)} = x_v$ for $v \in \mathcal{V}^{(L)}$:*

$$\bar{q}_v^{(\ell)} = \bar{f}_{\iota(v)}^{(\ell)}(\bar{h}_v^{(\ell)}) \in \mathbb{R}^S, \qquad\qquad \ell \in [L-1], \quad v \in \mathcal{V}^{(\ell)},$$
$$\bar{h}_v^{(\ell-1)} = \text{normalize}\Big(\sum_{v' \in \mathcal{C}(v)} \bar{q}_{v'}^{(\ell)}\Big) \in \mathbb{R}^S, \qquad \ell \in [L-1], \quad v \in \mathcal{V}^{(\ell-1)}. \tag{A-MP-CLS}$$

*Taking $h_r^{(0)} \in \mathbb{R}^S$ to be as defined in Eq. (MP-CLS) and $\bar{h}_r^{(0)} \in \mathbb{R}^S$ to be as defined in Eq. (A-MP-CLS), we have*

$$\|h_r^{(0)} - \bar{h}_r^{(0)}\|_\infty \leq \delta \times \prod_{1 \leq \ell \leq L}(2m^{(\ell)} + 1).$$

*Proof of Lemma 5.* We prove this lemma by induction, aiming to show that for any $\ell \in [L-1]$ we have

$$\|\bar{h}_v^{(\ell)} - h_v^{(\ell)}\|_\infty \leq 2m^{(\ell+1)} \prod_{k=\ell+2}^{L}(2m^{(k)} + 1)\delta, \quad \forall v \in \mathcal{V}^{(\ell)}. \tag{27}$$

To prove the formula for $\ell = L - 1$, since $h_v^{(L)} = \bar{h}_v^{(L)}$, by Eq. (26), we get

$$\|\bar{q}_v^{(L)} - q_v^{(L)}\|_\infty \leq \delta, \quad \forall v \in \mathcal{V}^{(L)}.$$

By Lemma 7, we get

$$\|h_v^{(L-1)} - \bar{h}_v^{(L-1)}\|_\infty \leq 2\Big\|\sum_{v' \in \mathcal{C}(v)}(\bar{q}_{v'}^{(L)} - q_{v'}^{(L)})\Big\|_\infty \leq 2m^{(L)}\delta, \quad \forall v \in \mathcal{V}^{(L-1)}.$$

This proves the formula (27) for $\ell = L - 1$.

Assuming that (27) holds at the layer $\ell$, by the update formula, we have

$$\|\bar{q}_v^{(\ell)} - q_v^{(\ell)}\|_\infty = \|\bar{f}_{\iota(v)}^{(\ell)}(\bar{h}_v^{(\ell)}) - f_{\iota(v)}^{(\ell)}(h_v^{(\ell)})\|_\infty$$
$$\leq \|\bar{f}_{\iota(v)}^{(\ell)}(\bar{h}_v^{(\ell)}) - f_{\iota(v)}^{(\ell)}(\bar{h}_v^{(\ell)})\|_\infty + \|f_{\iota(v)}^{(\ell)}(h_v^{(\ell)}) - f_{\iota(v)}^{(\ell)}(\bar{h}_v^{(\ell)})\|_\infty$$
$$\leq \delta + 2m^{(\ell+1)} \prod_{k=\ell+2}^{L}(2m^{(k)} + 1)\delta \leq \prod_{k=\ell+1}^{L}(2m^{(k)} + 1)\delta,$$

where the middle inequality is by the assumption of $f_\iota^{(\ell)}$ and by Lemma 6. By Lemma 7, we get

$$\|\bar{h}_v^{(\ell-1)} - h_v^{(\ell-1)}\|_\infty \leq 2m^{(\ell)} \prod_{k=\ell+1}^{L}(2m^{(k)} + 1)\delta, \quad \forall v \in \mathcal{V}^{(\ell)}.$$

This proves Lemma 5 by the induction argument. $\qquad\square$

**Lemma 6** (Non-expansiveness of log-sum-exponential). *For $h \in \mathbb{R}^S$ and $\Psi \in \mathbb{R}^{S \times S}$, define $f(h) \in \mathbb{R}^S$ by*

$$f(h)_i = \log \sum_{j=1}^{S} \Psi_{ij} \exp(h_j), \quad \forall i \in [S].$$

*Then for $h_1, h_2 \in \mathbb{R}^S$, we have*

$$\|f(h_1) - f(h_2)\|_\infty \leq \|h_1 - h_2\|_\infty.$$

*Proof of Lemma 6.* Fix $i \in [S]$. We have

$$\nabla_h f(h)_i = \Big(\frac{\Psi_{ij}\exp(h_j)}{\sum_{k \in [S]} \Psi_{ik}\exp(h_k)}\Big)_{j \in [S]},$$

so that $\|\nabla_h f(h)_i\|_1 = 1$. By intermediate value theorem, we have

$$|f(h_1)_i - f(h_2)_i| = |\nabla_h f(\xi)_i^\mathbb{T}(h_1 - h_2)| \leq \|\nabla_h f(\xi)_i^\mathbb{T}\|_1 \|h_1 - h_2\|_\infty = \|h_1 - h_2\|_\infty.$$

This proves Lemma 6. $\qquad\square$

**Lemma 7** (Lipschitzness of the normalization operator). *For $h \in \mathbb{R}^S$, define* $\mathrm{normalize}(h) \in \mathbb{R}^S$ *by*

$$\mathrm{normalize}(h)_i = h_i - \max_j h_j, \quad \forall i \in [S].$$

*Then for $h_1, h_2 \in \mathbb{R}^S$, we have*

$$\|\mathrm{normalize}(h_1) - \mathrm{normalize}(h_2)\|_\infty \le 2\|h_1 - h_2\|_\infty.$$

*Proof of Lemma 7.* Note we have the following inequality

$$|\max_j h_{1,j} - \max_j h_{2,j}| \le \|h_1 - h_2\|_\infty,$$

so that

$$|\mathrm{normalize}(h_1)_i - \mathrm{normalize}(h_2)_i| \le \|h_1 - h_2\|_\infty + |\max_j h_{1,j} - \max_j h_{2,j}| \le 2\|h_1 - h_2\|_\infty.$$

This completes the proof of Lemma 7. $\qquad\square$

**Lemma 8** (ReLU approximation of the exponential function). *For any $\delta > 0$, take $M = \lceil 1/\delta \rceil + 1 \in \mathbb{N}$. Then there exists $\{(a_j, w_j, b_j)\}_{j \in [M]}$ with*

$$\sup_j |a_j| \le 2, \quad \sup_j |w_j| \le 1, \quad \sup_j |b_j| \le \log M, \tag{28}$$

*such that defining $\exp_\delta : \mathbb{R} \to \mathbb{R}$ by*

$$\exp_\delta(x) = \sum_{j=1}^M a_j \cdot \mathrm{ReLU}(w_j x + b_j),$$

*we have $\exp_\delta$ is non-decreasing on $(-\infty, 0]$, and*

$$\sup_{x \in (-\infty, 0]} |\exp(x) - \exp_\delta(x)| \le \delta, \quad \exp_\delta(0) = 1.$$

*Proof of Lemma 8.* Define $e_j = j/(M-1)$, $b_j = -\log(e_j)$ for $j \in [M-1]$, $a_1 = (e_2 - e_1)/(b_2 - b_1)$ and $a_j = (e_{j+1} - e_j)/(b_{j+1} - b_j) - (e_j - e_{j-1})/(b_j - b_{j-1})$ for $2 \le j \le M-2$. Furthermore, define

$$\exp_\delta(x) = \sum_{j=1}^{M-2} a_j \mathrm{ReLU}(x + b_j) + \mathrm{ReLU}(-x + e_1) - \mathrm{ReLU}(-x).$$

Then we have $\exp_\delta(-b_j) = e_j$ for $j \in [M-1]$, and $\exp_\delta$ is piece-wise linear and non-decreasing on $(-\infty, 0]$. Note that we also have $\exp(-b_j) = e_j$ for $j \in [M-1]$, and $\exp$ is increasing on $(-\infty, 0]$. This proves that $\sup_{x \in (-\infty, 0]} |\exp(x) - \exp_\delta(x)| \le \delta$. Furthermore, it is easy to see that $\exp_\delta(0) = 1$ and $\exp_\delta$ is non-decreasing on $(-\infty, 0]$. Finally, since $\exp$ is 1-Lipschitz, it is easy to see that $\sup_{j \in [M]} |a_j| \le 2$. It is also easy to see the other parts of Eq. (28) are satisfied, and this proves Lemma 8. $\qquad\square$

**Lemma 9** (ReLU approximation of the logarithm function). *For any $A > 0$, $\delta > 0$, take $M = \lceil 2A/\delta \rceil + 1 \in \mathbb{N}$. Then there exists $\{(a_j, w_j, b_j)\}_{j \in [M]}$ with*

$$\sup_j |a_j| \le 2A, \quad \sup_j |w_j| \le 1, \quad \sup_j |b_j| \le A, \tag{29}$$

*such that defining $\log_\delta : \mathbb{R} \to \mathbb{R}$ by*

$$\log_\delta(x) = \sum_{j=1}^M a_j \cdot \mathrm{ReLU}(w_j x + b_j),$$

*we have $\log_\delta$ is non-decreasing on $[1/A, A]$, and*

$$\sup_{x \in [1/A, A]} |\log(x) - \log_\delta(x)| \le \delta.$$

*Proof of Lemma 9.* The proof of Lemma 9 is similar to Lemma 8. $\qquad\square$

**Lemma 10** (ReLU approximation of indicator function). *Define*

$$\mathrm{Ind}(x) = 2\mathrm{ReLU}(x - 1/2) + 2\mathrm{ReLU}(x + 1/2) - 4\mathrm{ReLU}(x),$$

*we have*

$$1(x = j) = \mathrm{Ind}(x - j), \quad \forall j, x \in \mathbb{Z}.$$

*Proof of Lemma 10.* The lemma holds by direct calculation. □

**Lemma 11** (Log-sum-exponential approximation). *Assume $\log_{\delta_1} : \mathbb{R} \to \mathbb{R}$ and $\exp_{\delta_2} : \mathbb{R} \to \mathbb{R}$ are such that,*

$$\sup_{x \in [1/K, SK]} |\log(x) - \log_{\delta_1}(x)| \le \delta_1, \qquad \sup_{x \in (-\infty, 0]} |\exp(x) - \exp_{\delta_2}(x)| \le \delta_2,$$

$$\exp_{\delta_2}(0) = 1, \qquad \exp_{\delta_2} \text{ is non-decreasing on } (-\infty, 0].$$

*Assume that $1/K \le \min_{ij} \Psi_{ij} \le \max_{ij} \Psi_{ij} \le K$. Define $f(h), f_{\delta_1, \delta_2}(h) \in \mathbb{R}^S$ by*

$$f(h)_i = \log \sum_{j=1}^S \Psi_{ij} \exp(h_j), \quad f_{\delta_1, \delta_2}(h)_i = \log_{\delta_1} \sum_{j=1}^S \Psi_{ij} \exp_{\delta_2}(h_j), \quad \forall i \in [S].$$

*Then we have*

$$\sup_{\max_i h_i = 0} \|f(h) - f_{\delta_1, \delta_2}(h)\|_\infty \le \delta_1 + SK^2 \delta_2.$$

*Proof of Lemma 11.* We have

$$|f(h)_i - f_{\delta_1, \delta_2}(h)_i| = |\log \langle \Psi_{i:}, \exp(h) \rangle - \log_{\delta_1} \langle \Psi_{i:}, \exp_{\delta_2}(h) \rangle|$$
$$\le |\log \langle \Psi_{i:}, \exp_{\delta_2}(h) \rangle - \log_{\delta_1} \langle \Psi_{i:}, \exp_{\delta_2}(h) \rangle| + |\log \langle \Psi_{i:}, \exp(h) \rangle - \log \langle \Psi_{i:}, \exp_{\delta_2}(h) \rangle|.$$

For the first term, since $\exp_{\delta_2}(h) \le 1$ for all $h \le 0$, we have $\langle \Psi_{i:}, \exp_{\delta_2}(h) \rangle \le S \max_{ij} \Psi_{ij} \le SK$. Furthermore, since $\max_i h_i = 0$ and $\exp_{\delta_2}(0) = 1$, we have $\langle \Psi_{i:}, \exp_{\delta_2}(h) \rangle \ge \min_{ij} \Psi_{ij} \ge 1/K$. As a consequence, by assumption, we have

$$|\log \langle \Psi_{i:}, \exp_{\delta_2}(h) \rangle - \log_{\delta_1} \langle \Psi_{i:}, \exp_{\delta_2}(h) \rangle| \le \delta_1.$$

For the second term, since both $\langle \Psi_{i:}, \exp(h) \rangle$ and $\langle \Psi_{i:}, \exp_{\delta_2}(h) \rangle$ are within $[1/K, SK]$ on which $\log$ function has Lipschitz constant $K$, we have

$$|\log \langle \Psi_{i:}, \exp(h) \rangle - \log \langle \Psi_{i:}, \exp_{\delta_2}(h) \rangle| \le SK \max_{ij} \Psi_{ij} \cdot \|\exp(h) - \exp_{\delta_2}(h)\|_\infty \le SK^2 \delta_2.$$

This finishes the proof of Lemma 11. □

**Lemma 12** (Log-Psi approximation). *Assume $\log_\delta : \mathbb{R} \to \mathbb{R}$ is such that,*

$$\sup_{x \in [1/K, K]} |\log(x) - \log_\delta(x)| \le \delta.$$

*Assume that $1/K \le \min_{ij} \Psi_{ij} \le \max_{ij} \Psi_{ij} \le K$. For $x \in [S]$, define $f(x), f_\delta(x) \in \mathbb{R}^S$ by*

$$f(x)_i = \log \sum_{j=1}^S \Psi_{ij} 1(x = j), \quad f_\delta(x)_i = \log_\delta \sum_{j=1}^S \Psi_{ij} 1(x = j), \quad \forall i \in [S].$$

*Then we have*

$$\sup_{x \in [S]} \|f(x) - f_\delta(x)\|_\infty \le \delta.$$

*Proof of Lemma 12.* For any fixed $x \in [S]$ and $i \in [S]$, we have

$$|f(x)_i - f_\delta(x)_i| = |\log \Psi_{ix} - \log_\delta \Psi_{ix}| \le \delta,$$

where the last inequality is by assumption. This proves Lemma 12. □

**Lemma 13** (ReLU approximation of log-sum-exponential). *Assume that* $1/K \leq \min_{ij} \Psi_{ij} \leq \max_{ij} \Psi_{ij} \leq K$. *For any* $\delta > 0$, *take* $M_1 = \lceil 2SK/\delta \rceil + 1$ *and* $M_2 = \lceil 2SK^2/\delta \rceil + 1$. *Then there exists* $\{(a_j, w_j, b_j)\}_{j \in [M_1]}$ *and* $\{(\bar{a}_j, \bar{w}_j, \bar{b}_j)\}_{j \in [M_2]}$ *with*

$$\sup_j |a_j| \leq 2SK, \quad \sup_j |w_j| \leq 1, \quad \sup_j |b_j| \leq SK, \quad \sup_j |\bar{a}_j| \leq 2, \quad \sup_j |\bar{w}_j| \leq 1, \quad \sup_j |\bar{b}_j| \leq \log(4SK^2/\delta),$$

*such that defining*

$$\log_{\delta\star}(x) = \sum_{j=1}^{M_1} a_j \cdot \mathrm{ReLU}(w_j x + b_j), \quad \exp_{\delta\star}(x) = \sum_{j=1}^{M_2} \bar{a}_j \cdot \mathrm{ReLU}(\bar{w}_j x + \bar{b}_j),$$

*and defining* $f(h), f_\delta(h) \in \mathbb{R}^S$ *by*

$$f(h)_i = \log \sum_{j=1}^S \Psi_{ij} \exp(h_j), \quad f_\delta(h)_i = \log_{\delta\star} \sum_{j=1}^S \Psi_{ij} \exp_{\delta\star}(h_j), \quad \forall i \in [S],$$

*we have*

$$\sup_{\max_i h_i = 0} \|f(h) - f_\delta(h)\|_\infty \leq \delta.$$

*Proof of Lemma 13.* Lemma 9 implies that taking $M_1 = \lceil 2SK/\delta \rceil + 1$, there exists $\{(a_j, w_j, b_j)\}_{j \in [M_1]}$ with

$$\sup_j |a_j| \leq 2SK, \quad \sup_j |w_j| \leq 1, \quad \sup_j |b_j| \leq SK,$$

such that defining

$$\log_{\delta\star}(x) = \sum_{j=1}^{M_1} a_j \cdot \mathrm{ReLU}(w_j x + b_j),$$

we have

$$\sup_{1/(SK) \leq x \leq SK} |\log(x) - \log_{\delta\star}(x)| \leq \delta/2.$$

Lemma 8 implies that taking $M_2 = \lceil 2SK^2/\delta \rceil + 1$, there exists $\{(\bar{a}_j, \bar{w}_j, \bar{b}_j)\}_{j \in [M_2]}$ with

$$\sup_j |\bar{a}_j| \leq 2, \quad \sup_j |\bar{w}_j| \leq 1, \quad \sup_j |\bar{b}_j| \leq \log(4SK^2/\delta),$$

such that defining

$$\exp_{\delta\star}(x) = \sum_{j=1}^{M_2} \bar{a}_j \cdot \mathrm{ReLU}(\bar{w}_j x + \bar{b}_j),$$

we have $\exp_{\delta\star}$ is non-decreasing on $(-\infty, 0]$, and

$$\sup_{x \in (-\infty, 0]} |\exp(x) - \exp_{\delta\star}(x)| \leq \delta/(2SK^2), \quad \exp_{\delta\star}(0) = 1.$$

As a consequence, the condition of Lemma 11 is satisfied with $\delta_1 = \delta/2$ and $\delta_2 = \delta/(2SK^2)$, so that we have

$$\sup_{\max_i h_i = 0} \|f(h) - f_\delta(h)\|_\infty \leq \delta_1 + SK^2\delta_2 = \delta.$$

This finishes the proof of Lemma 13. $\qquad\qquad\qquad\qquad\qquad\qquad\qquad\qquad\qquad\qquad\qquad\square$

**Lemma 14** (Existence of ReLU network approximating log-sum-exponential). *Let* $f_\delta$ *be the function as defined in Lemma 13. Then there exists a two-hidden-layer neural network*

$$\mathrm{NN}_{W_1, W_2, W_3}(h) = W_1 \cdot \mathrm{ReLU}(W_2 \cdot \mathrm{ReLU}(W_3 \cdot [h; 1])),$$

*with* $W_1 \in \mathbb{R}^{S \times SM_1}$, $W_2 \in \mathbb{R}^{SM_1 \times (SM_2+1)}$, $W_3 \in \mathbb{R}^{(SM_2+1) \times (S+1)}$, *and*

$$\|W_1\|_{\max} \leq 2SK, \quad \|W_2\|_{\max} \leq \mathrm{Poly}(SKM_1M_2), \quad \|W_3\|_{\max} \leq \log(4SK^2/\delta).$$

*such that*

$$\mathrm{NN}_{W_1, W_2, W_3}(h) = f_\delta(h), \quad \forall h \in \mathbb{R}^S \text{ such that } \max_j h_j = 0.$$

*Proof of Lemma 14.* Define

$$\overline{W}_2 = \begin{bmatrix} \bar{a}_{1:M_2}^{\mathbb{T}} & 0 & \cdots & 0 & 0 \\ 0 & \bar{a}_{1:M_2}^{\mathbb{T}} & \cdots & 0 & 0 \\ \cdots & \cdots & \cdots & \cdots & \cdots \\ 0 & 0 & \cdots & \bar{a}_{1:M_2}^{\mathbb{T}} & 0 \\ 0 & 0 & \cdots & 0 & 1 \end{bmatrix} \in \mathbb{R}^{(S+1)\times(SM_2+1)},$$

$$W_3 = \begin{bmatrix} \bar{w}_{1:M_2} & 0 & \cdots & 0 & \bar{b}_{1:M_2} \\ 0 & \bar{w}_{1:M_2} & \cdots & 0 & \bar{b}_{1:M_2} \\ \cdots & \cdots & \cdots & \cdots & \cdots \\ 0 & 0 & \cdots & \bar{w}_{1:M_2} & \bar{b}_{1:M_2} \\ 0 & 0 & \cdots & 0 & 1 \end{bmatrix} \in \mathbb{R}^{(SM_2+1)\times(S+1)},$$

then we have
$$[\exp_{\delta\star}(h); 1] = \overline{W}_2 \cdot \mathrm{ReLU}(W_3 \cdot [h; 1]).$$

Define

$$W_1 = \begin{bmatrix} a_{1:M_1}^{\mathbb{T}} & 0 & \cdots & 0 \\ 0 & a_{1:M_1}^{\mathbb{T}} & \cdots & 0 \\ \cdots & \cdots & \cdots & \cdots \\ 0 & 0 & \cdots & a_{1:M_1}^{\mathbb{T}} \end{bmatrix} \in \mathbb{R}^{S\times SM_1},$$

$$\tilde{W}_2 = \begin{bmatrix} w_{1:M_1} & 0 & \cdots & 0 & b_{1:M_1} \\ 0 & w_{1:M_1} & \cdots & 0 & b_{1:M_1} \\ \cdots & \cdots & \cdots & \cdots & \cdots \\ 0 & 0 & \cdots & w_{1:M_1} & \bar{b}_{1:M_1} \end{bmatrix} \in \mathbb{R}^{SM_1\times(S+1)},$$

then we have
$$\log_{\delta\star}(h) = W_1 \cdot \mathrm{ReLU}(\tilde{W}_2 \cdot [h; 1]).$$

As a consequence, we have

$$f_\star(h) = \log_{\delta\star} \Psi \exp_{\delta\star}(h) = W_1 \cdot \mathrm{ReLU}(\tilde{W}_2 \cdot \mathrm{diag}(\Psi, 1) \cdot \overline{W}_2 \cdot \mathrm{ReLU}(W_3 \cdot [h; 1])) = W_1 \cdot \mathrm{ReLU}(W_2 \cdot \mathrm{ReLU}(W_3 \cdot [h; 1])),$$

where we define $W_2 = \tilde{W}_2 \cdot \mathrm{diag}(\Psi, 1) \cdot \overline{W}_2$. It is also direct to upper bound $\|W_1\|_{\max}, \|W_2\|_{\max}$, and $\|W_3\|_{\max}$. This finishes the proof of Lemma 14. $\qquad\square$

**Lemma 15** (ReLU approximation of log-Psi). *Assume that $1/K \leq \min_{ij} \Psi_{ij} \leq \max_{ij} \Psi_{ij} \leq K$. For any $\delta > 0$, take $M_1 = \lceil K/\delta \rceil + 1$ and $M_2 = 3$. Then there exists $\{(a_j, w_j, b_j)\}_{j\in[M_1]}$ and $\{(\bar{a}_j, \bar{w}_j, \bar{b}_j)\}_{j\in[M_2]}$ with*

$$\sup_j |a_j| \leq 2K, \quad \sup_j |w_j| \leq 1, \quad \sup_j |b_j| \leq SK, \quad \sup_j |\bar{a}_j| \leq 4, \quad \sup_j |\bar{w}_j| \leq 1, \quad \sup_j |\bar{b}_j| \leq 1,$$

*such that defining*

$$\log_\delta(x) = \sum_{j=1}^{M_1} a_j \cdot \mathrm{ReLU}(w_j x + b_j), \quad \mathrm{Ind}(x) = \sum_{j=1}^{M_2} \bar{a}_j \cdot \mathrm{ReLU}(w_j x + b_j),$$

*and defining $f(x), f_\delta(x) \in \mathbb{R}^S$ by*

$$f(x)_i = \log \sum_{j=1}^{S} \Psi_{ij} 1(x = j), \quad f_\delta(x)_i = \log_\delta \sum_{j=1}^{S} \Psi_{ij} \mathrm{Ind}(x - j), \quad \forall i \in [S],$$

*we have*

$$\sup_{x\in[S]} \|f(x) - f_\delta(x)\|_\infty \leq \delta.$$

*Proof of Lemma 15.* The proof of the lemma is similar to the proof of Lemma 13, using a combination of Lemma 10, 9, and 12. $\qquad\square$

**Lemma 16** (Existence of ReLU network approximating log-Psi). *Let $f_\delta$ be the function as defined in Lemma 15. Then there exists a two-hidden-layer neural network*

$$\mathrm{NN}_{W_1, W_2, W_3}(x) = W_1 \cdot \mathrm{ReLU}(W_2 \cdot \mathrm{ReLU}(W_3 \cdot [x; 1])),$$

*with $W_1 \in \mathbb{R}^{S\times SM_1}$, $W_2 \in \mathbb{R}^{SM_1\times(SM_2+1)}$, $W_3 \in \mathbb{R}^{(SM_2+1)\times(S+1)}$, and*

$$\|W_1\|_{\max} \leq 2K, \quad \|W_2\|_{\max} \leq \mathrm{Poly}(SKM_1M_2), \quad \|W_3\|_{\max} \leq 1.$$

*such that*

$$\mathrm{NN}_{W_1, W_2, W_3}(x) = f_\delta(x), \quad \forall x \in [S].$$

*Proof of Lemma 16.* The proof of Lemma 16 is similar to the proof of Lemma 14. $\qquad\square$

# E    PROOF OF THEOREM 1

*Proof of Theorem 1.* By Lemma 17, we have the error decomposition

$$\mathsf{D}_2^2(\mu_{\mathrm{NN}}^{\widehat{\boldsymbol{W}}}, \mu_\star) \leq \inf_{\boldsymbol{W} \in \mathcal{W}} \mathsf{D}_2^2(\mu_{\mathrm{NN}}^{\boldsymbol{W}}, \mu_\star) + 2 \cdot \sup_{\boldsymbol{W} \in \mathcal{W}} \left| \widehat{\mathsf{R}}(\mu_{\mathrm{NN}}^{\boldsymbol{W}}) - \mathsf{R}(\mu_{\mathrm{NN}}^{\boldsymbol{W}}) \right|.$$

To control the first term (the approximation error), by Theorem 4, there exists $\boldsymbol{W} \in \mathcal{W}_{d,\underline{m},L,S,D,B}$ as in Eq. (9) with norm bound $B = \mathrm{Poly}(d, S, K, 3^L, D)$, such that defining $\mu_{\mathrm{NN}}^{\boldsymbol{W}}$ as in Eq. (ConvNet), we have

$$\max_{y \in [S], \boldsymbol{x} \in [S]^d} \left| \log \mu_\star(y|\boldsymbol{x}) - \log \mu_{\mathrm{NN}}^{\boldsymbol{W}}(y|\boldsymbol{x}) \right| \leq C \cdot \frac{S^2 K^2 d \cdot 3^L}{D}.$$

Furthermore, by Lemma 4, when $D \geq S^2 K^2 d \cdot 3^L$, we have

$$\inf_{\boldsymbol{W} \in \mathcal{W}} \mathsf{D}_2^2(\mu_{\mathrm{NN}}^{\boldsymbol{W}}, \mu_\star) \leq C \Big( e^{\frac{S^2 K^2 d \cdot 3^L}{D}} - 1 \Big)^2 \leq C \frac{S^4 K^4 d^2 \cdot 3^{2L}}{D^2}.$$

To control the second term (the generalization error), by Proposition 7, with probability at least $1 - \eta$, we have

$$\sup_{\boldsymbol{W} \in \mathcal{W}_{d,\underline{m},L,S,D,B}} \left| \widehat{\mathsf{R}}(\mu_{\mathrm{NN}}^{\boldsymbol{W}}) - \mathsf{R}(\mu_{\mathrm{NN}}^{\boldsymbol{W}}) \right| \leq C \cdot \sqrt{\frac{LD(D + 2S + 1)\|m\|_1 \log(d\|m\|_1 DSB \cdot 3^L) + \log(1/\eta)}{n}}.$$

Combining the above two equations proves Theorem 1.                                     □

## E.1    ERROR DECOMPOSITION

**Lemma 17.** *Consider the setting of Theorem 1. We have decomposition*

$$\mathsf{D}_2^2(\mu_{\mathrm{NN}}^{\widehat{\boldsymbol{W}}}, \mu_\star) \leq \inf_{\boldsymbol{W} \in \mathcal{W}} \mathsf{D}_2^2(\mu_{\mathrm{NN}}^{\boldsymbol{W}}, \mu_\star) + 2 \cdot \sup_{\boldsymbol{W} \in \mathcal{W}} \left| \widehat{\mathsf{R}}(\mu_{\mathrm{NN}}^{\boldsymbol{W}}) - \mathsf{R}(\mu_{\mathrm{NN}}^{\boldsymbol{W}}) \right|.$$

*Proof of Lemma 17.* We have that for any conditional distribution $\mu_1(\cdot|\cdot)$, there is decomposition

$$\mathsf{D}_2^2(\mu_1, \mu_\star) = \mathbb{E}_{\boldsymbol{x} \sim \mu_\star} \Big[ \sum_{s=1}^{S} \Big( \mu_1(s|\boldsymbol{x}) - \mu_\star(s|\boldsymbol{x}) \Big)^2 \Big]$$

$$= \mathbb{E}_{(\boldsymbol{x},y) \sim \mu_\star} \Big[ \sum_{s=1}^{S} (\mu_1(s|\boldsymbol{x}) - \mathbf{1}\{y = s\})^2 \Big] - \mathbb{E}_{(\boldsymbol{x},y) \sim \mu_\star} \Big[ \sum_{s=1}^{S} (\mu_\star(s|\boldsymbol{x}) - \mathbf{1}\{y = s\})^2 \Big] = \mathsf{R}(\mu_1) - \mathsf{R}(\mu_\star).$$

Define

$$\boldsymbol{W}_\star = \arg \min_{\boldsymbol{W} \in \mathcal{W}} \mathsf{R}(\mu_{\mathrm{NN}}^{\boldsymbol{W}}) = \arg \min_{\boldsymbol{W} \in \mathcal{W}} \mathsf{D}_2^2(\mu_{\mathrm{NN}}^{\boldsymbol{W}}, \mu_\star).$$

Then we have

$$\mathsf{D}_2^2(\mu_{\mathrm{NN}}^{\widehat{\boldsymbol{W}}}, \mu_\star) = \mathsf{R}(\mu_{\mathrm{NN}}^{\widehat{\boldsymbol{W}}}) - \mathsf{R}(\mu_\star)$$

$$= \mathsf{R}(\mu_{\mathrm{NN}}^{\widehat{\boldsymbol{W}}}) - \widehat{\mathsf{R}}(\mu_{\mathrm{NN}}^{\widehat{\boldsymbol{W}}}) + \widehat{\mathsf{R}}(\mu_{\mathrm{NN}}^{\widehat{\boldsymbol{W}}}) - \widehat{\mathsf{R}}(\mu_{\mathrm{NN}}^{\boldsymbol{W}_\star}) + \widehat{\mathsf{R}}(\mu_{\mathrm{NN}}^{\boldsymbol{W}_\star}) - \mathsf{R}(\mu_{\mathrm{NN}}^{\boldsymbol{W}_\star}) + \mathsf{R}(\mu_{\mathrm{NN}}^{\boldsymbol{W}_\star}) - \mathsf{R}(\mu_\star)$$

$$\leq 2 \cdot \sup_{\boldsymbol{W} \in \mathcal{W}} \left| \mathsf{R}(\mu_{\mathrm{NN}}^{\boldsymbol{W}}) - \widehat{\mathsf{R}}(\mu_{\mathrm{NN}}^{\boldsymbol{W}}) \right| + \mathsf{D}_2^2(\mu_{\mathrm{NN}}^{\boldsymbol{W}_\star}, \mu_\star)$$

This proves Lemma 17.                                     □

## E.2    RESULTS ON GENERALIZATION

**Proposition 7** (Generalization error of the classification problem)**.** *Let $\mathcal{W}_{d,\underline{m},L,S,D,B}$ be the set defined as in Eq. (9). Then, with probability at least $1 - \eta$, we have*

$$\sup_{\boldsymbol{W} \in \mathcal{W}_{d,\underline{m},L,S,D,B}} \left| \widehat{\mathsf{R}}(\mu_{\mathrm{NN}}^{\boldsymbol{W}}) - \mathsf{R}(\mu_{\mathrm{NN}}^{\boldsymbol{W}}) \right| \leq C \cdot \sqrt{\frac{LD(D + 2S + 1)\|m\|_1 \log(d\|m\|_1 DSB \cdot 3^L) + \log(1/\eta)}{n}}.$$

*Proof of Proposition 7.* In Lemma 3, we can take $z = (y, \boldsymbol{x})$, $w = \boldsymbol{W}$, $\Theta = \mathcal{W}_{d,\underline{m},L,S,D,B}$, $\rho(w, w') = \|\boldsymbol{W} - \boldsymbol{W}'\|$, and $f(z_i; w) = \text{loss}(y, \mu_{\text{NN}}^{\boldsymbol{W}}(\cdot | \boldsymbol{x}))$. Therefore, to show Proposition 7, we just need to apply Lemma 3 by checking (a), (b), (c).

**Check (a).** We note that the index set $\Theta := \mathcal{W}_{d,\underline{m},L,S,D,B}$ equipped with $\rho(w, w') := \|\boldsymbol{W} - \boldsymbol{W}'\|$ has diameter $B_{\text{p}} := 2B$. Further note that $\mathcal{W}_{d,\underline{m},L,S,D,B}$ has a dimension bounded by $d_{\text{p}} := D(D + 2S + 1)\|\underline{m}\|_1$. According to Example 5.8 of (Wainwright, 2019), it holds that $\log N(\Delta; \mathcal{W}_{d,\underline{m},L,S,D,B}, \|\cdot\|) \leq d_{\text{p}} \cdot \log(1 + 2r/\Delta)$ for any $0 < \Delta \leq 2r$. This verifies (a).

**Check (b).** Since $f(z_i; w) = \text{loss}(y, \mu_{\text{NN}}^{\boldsymbol{W}}(\cdot | \boldsymbol{x}))$ is 2-bounded. As a consequence, $f(z, w) - \mathbb{E}_z[f(z, w)]$ is a sub-Gaussian random variable with the sub-Gaussian parameter to be a universal constant.

**Check (c).** Lemma 20 implies that
$$|f(z; w_1) - f(z; w_2)| \leq \sigma' \cdot \|\boldsymbol{W}_1 - \boldsymbol{W}_2\|, \quad \sigma' := 12\|\underline{m}\|_1 (3B^3)^L \cdot d \cdot S^{3/2} \cdot (S + D).$$
As a consequence, $f(z; w_1) - f(z; w_2)$ is $\sigma'\rho(w_1, w_2) = \sigma'\|\boldsymbol{W}_1 - \boldsymbol{W}_2\|$ sub-Gaussian.

Therefore, we apply Lemma 3 to conclude the proof of Proposition 7. $\qquad\square$

### E.3 AUXILLARY LEMMAS

**Lemma 18** (Norm bound in the chain rule in classification settings)**.** *Consider the ConvNet as in Eq. (ConvNet). Assume that $\|\boldsymbol{W}\| \leq B$. Then for any $\ell$, $v$, $\iota$, and $\star \in [3]$, we have*

$$\|\mathsf{h}_v^{(L)}\|_2 \leq S^{3/2},$$
$$\|\mathsf{q}_v^{(\ell)}\|_2 \leq B^3 \cdot (\|\mathsf{h}_v^{(\ell)}\|_2 + 1),$$
$$\|\mathsf{h}_v^{(\ell-1)}\|_2 \leq 2m^{(\ell)} \cdot \max_{v' \in \mathcal{C}(v)} \|\mathsf{q}_{v'}^{(\ell)}\|_2,$$
$$\max_{i \in [S]} \|\nabla_{W_{\star,\iota}^{(k)}} \mathsf{q}_{v,i}^{(\ell)}\|_{\text{op}} \leq B^3 \cdot \max_{i \in [S]} \|\nabla_{W_{\star,\iota}^{(k)}} \mathsf{h}_{v,i}^{(\ell)}\|_{\text{op}}, \qquad\qquad \forall k \geq \ell + 1,$$
$$\max_{i \in [S]} \|\nabla_{W_{\star,\iota}^{(\ell)}} \mathsf{q}_{v,i}^{(\ell)}\|_{\text{op}} \leq B^2 \cdot \|\mathsf{h}_v^{(\ell)}\|_2,$$
$$\max_{i \in [S]} \|\nabla_{W_{\star,\iota}^{(k)}} \mathsf{h}_{v,i}^{(\ell-1)}\|_{\text{op}} \leq 2m^{(\ell)} \cdot \max_{v' \in \mathcal{C}(v)} \max_{i \in [S]} \|\nabla_{W_{\star,\iota}^{(k)}} \mathsf{q}_{v',i}^{(\ell)}\|_{\text{op}}, \qquad \forall k \geq \ell,$$
$$\max_{i \in [S]} \|\nabla_{W_{\star,\iota}^{(\ell)}} \text{softmax}(\mathsf{h}_{\text{r}}^{(0)})_i\|_{\text{op}} \leq \max_{i \in [S]} \|\nabla_{W_{\star,\iota}^{(\ell)}} \mathsf{h}_{\text{r},i}^{(0)}\|_{\text{op}}.$$

*Proof of Lemma 18.* The proof of the lemma uses the chain rule, the 1-Lipschitzness of ReLU, the 2-Lipschitzness of normalize, and the 1-Lipschitzness of softmax. $\qquad\square$

**Lemma 19.** *Consider the ConvNet as in Eq. (ConvNet). Assume that $\|\boldsymbol{W}\| \leq B$. Then we have*
$$\max_{\boldsymbol{x} \in [S]^d} \max_{\star \in [3]} \max_{\ell \in [L]} \max_{\iota \in [m^{(\ell)}]} \max_{i \in [S]} \|\nabla_{W_{\star,\iota}^{(\ell)}} \text{softmax}(\mathsf{h}_{\text{r}}^{(0)})_i\|_{\text{op}} \leq (3B^3)^L \cdot d \cdot S^{3/2}.$$

*Proof of Lemma 19.* This lemma is implied by Lemma 18 and an induction argument. $\qquad\square$

**Lemma 20.** *Consider the ConvNet as in Eq. (ConvNet). Assume that $\|\boldsymbol{W}\| \leq B$. Then we have*
$$\max_{y,\boldsymbol{x}} \left|\mu_{\text{NN}}^{\boldsymbol{W}}(y|\boldsymbol{x}) - \mu_{\text{NN}}^{\overline{\boldsymbol{W}}}(y|\boldsymbol{x})\right| \leq 3\|\underline{m}\|_1 (3B^3)^L \cdot d \cdot S^{3/2} \cdot (S + D) \cdot \|\boldsymbol{W} - \overline{\boldsymbol{W}}\|.$$
*Therefore, we have*
$$\left|\text{loss}(y, \mu_{\text{NN}}^{\boldsymbol{W}}(\cdot|\boldsymbol{x})) - \text{loss}(y, \mu_{\text{NN}}^{\overline{\boldsymbol{W}}}(\cdot|\boldsymbol{x}))\right| \leq 12\|\underline{m}\|_1 (3B^3)^L \cdot d \cdot S^{3/2} \cdot (S + D) \cdot \|\boldsymbol{W} - \overline{\boldsymbol{W}}\|.$$

*Proof of Lemma 20.* The first inequality is by the fact that
$$\max_{y,\boldsymbol{x}} \left|\mu_{\text{NN}}^{\boldsymbol{W}}(y|\boldsymbol{x}) - \mu_{\text{NN}}^{\overline{\boldsymbol{W}}}(y|\boldsymbol{x})\right|$$
$$\leq \sum_{\star \in [3]} \sum_{\ell \in [L]} \sum_{\iota \in [m^{(\ell)}]} \min\{\text{nrow}(W_{\star,\iota}^{(\ell)}), \text{ncol}(W_{\star,\iota}^{(\ell)})\} \|\nabla_{W_{\star,\iota}^{(\ell)}} \text{softmax}(\mathsf{h}_{\text{r}}^{(0)})_i\|_{\text{op}} \|W_{\star,\iota}^{(\ell)} - \overline{W}_{\star,\iota}^{(\ell)}\|_{\text{op}},$$

where we have used the inequality that $\mathrm{trace}(A^{\mathsf{T}}B) \leq \{\mathrm{nrow}(A), \mathrm{ncol}(A)\}\|A\|_{\mathrm{op}}\|B\|_{\mathrm{op}}$.

To prove the second equation, we have

$$\left|\mathrm{loss}(y, \mu_{\mathrm{NN}}^{\boldsymbol{W}}(\cdot|\boldsymbol{x})) - \mathrm{loss}(y, \mu_{\mathrm{NN}}^{\overline{\boldsymbol{W}}}(\cdot|\boldsymbol{x}))\right| = \left|\sum_{s=1}^{S}\left(1\{y=s\} - \mu_{\mathrm{NN}}^{\boldsymbol{W}}(s|\boldsymbol{x})\right)^2 - \sum_{s=1}^{S}\left(1\{y=s\} - \mu_{\mathrm{NN}}^{\overline{\boldsymbol{W}}}(s|\boldsymbol{x})\right)^2\right|$$

$$\leq \sum_{s=1}^{S}\left|1\{y=s\} - \mu_{\mathrm{NN}}^{\boldsymbol{W}}(s|\boldsymbol{x}) + 1\{y=s\} - \mu_{\mathrm{NN}}^{\overline{\boldsymbol{W}}}(s|\boldsymbol{x})\right| \cdot \left|\mu_{\mathrm{NN}}^{\boldsymbol{W}}(s|\boldsymbol{x}) - \mu_{\mathrm{NN}}^{\overline{\boldsymbol{W}}}(s|\boldsymbol{x})\right| \leq 4 \cdot \max_{s}\left|\mu_{\mathrm{NN}}^{\boldsymbol{W}}(s|\boldsymbol{x}) - \mu_{\mathrm{NN}}^{\overline{\boldsymbol{W}}}(s|\boldsymbol{x})\right|.$$

This completes the proof of Lemma 20. $\qquad\square$

## F    PROOF OF PROPOSITION 5

*Proof of Proposition 5.* By Eq. (MP-DNS) and (23), defining $\nu_{\downarrow,v}^{(\ell)}(\cdot) = \mathrm{softmax}(h_{\downarrow,v}^{(\ell)})$, then for $\ell \leq L-1$, we get

$$\nu_{\downarrow,v}^{(\ell)}(x_v^{(\ell)}) \propto \prod_{v' \in \mathcal{C}(v)}\left(\sum_{a \in [S]}\psi_{\iota(v')}^{(\ell+1)}(x_v^{(\ell)}, a)e^{(h_{\downarrow,v}^{(\ell+1)})_a}\right)$$

$$\propto \sum_{x_{\mathcal{C}(v)}^{(\ell+1)}}\prod_{v' \in \mathcal{C}(v)}\left(\psi_{\iota(v')}^{(\ell+1)}(x_v^{(\ell)}, x_{v'}^{(\ell+1)})\nu_{\downarrow,v'}^{(\ell+1)}(x_{v'}^{(\ell+1)})\right).$$

This coincides with the update rule of $\nu_{\downarrow,v}^{(\ell)}$ as in Eq. (BP-DNS).

Furthermore, defining $\nu_{\uparrow,v}^{(\ell)}(\cdot) = \mathrm{softmax}(b_{\uparrow,v}^{(\ell)} - h_{\downarrow,v}^{(\ell)})$, then for $\ell = 0, 1, \ldots, L$, we get

$$\nu_{\uparrow,v}^{(\ell)}(x_v^{(\ell)}) \propto \sum_{b \in [S]}\psi_{\iota(v)}^{(\ell+1)}(b, x_v^{(\ell)})\nu_{\uparrow,\mathrm{pa}(v)}^{(\ell+1)}(b)\prod_{v' \in \mathcal{N}(v)}\left(\sum_{a \in [S]}\psi_{\iota(v')}^{(\ell+1)}(b, a)e^{(h_{\downarrow,v}^{(\ell+1)})_a}\right)$$

$$\propto \sum_{x_{\mathrm{pa}(v)}^{(\ell-1)}, x_{\mathcal{N}(v)}^{(\ell)}}\psi^{(\ell)}(x_{\mathrm{pa}(v)}^{(\ell-1)}, x_{\mathcal{C}(\mathrm{pa}(v))}^{(\ell)})\nu_{\uparrow,\mathrm{pa}(v)}^{(\ell-1)}(x_{\mathrm{pa}(v)}^{(\ell-1)})\prod_{v' \in \mathcal{N}(v)}\nu_{\downarrow,v'}^{(\ell)}(x_{v'}^{(\ell)}).$$

This coincides with the update rule of $\nu_{\uparrow,v}^{(\ell)}$ as in Eq. (BP-DNS).

Finally, defining $\nu_v^{(L)}(\cdot) = \mathrm{softmax}(b_{\uparrow,v}^{(L)})$, we get

$$\nu_v^{(L)}(x_v^{(L)}) \propto \sum_{b \in [S]}\psi_{\iota(v)}^{(L)}(b, x_v^{(L)})\nu_{\uparrow,\mathrm{pa}(v)}^{(L)}(b)\prod_{v' \in \mathcal{N}(v)}\left(\sum_{a \in [S]}\psi_{\iota(v')}^{(L)}(b, a)e^{(h_{\downarrow,v}^{(L)})_a}\right) \times \nu_{\downarrow,v}^{(L)}(x_v^{(L)})$$

$$\propto \nu_{\uparrow,v}^{(L)}(x_v^{(L)})\psi^{(L+1)}(x_v^{(L)}, z_v).$$

This coincides with the formula of $\nu_v^{(L)}$ as in Eq. (BP-DNS).

This finishes the proof of Proposition 5. $\qquad\square$

## G    PROOF OF THEOREM 6

*Proof of Theorem 6.* By Lemma 13, take $M_1 = \lceil 2S^2Kd18^L/\delta\rceil + 1$ and $M_2 = \lceil 2S^2K^2d18^L/\delta\rceil + 1$. Then there exists $\{(a_j, w_j, b_j)\}_{j \in [M_1]}$ and $\{(\bar{a}_j, \bar{w}_j, \bar{b}_j)\}_{j \in [M_2]}$ with

$$\sup_j |a_j| \leq 2SK, \quad \sup_j |w_j| \leq 1, \quad \sup_j |b_j| \leq SK, \quad \sup_j |\bar{a}_j| \leq 2, \quad \sup_j |\bar{w}_j| \leq 1, \quad \sup_j |\bar{b}_j| \leq \log(4 \cdot 18^L S^2 dK^2/\delta),$$

such that defining

$$\log_{\delta\star}(x) = \sum_{j=1}^{M_1}a_j \cdot \mathrm{ReLU}(w_jx + b_j), \quad \exp_{\delta\star}(x) = \sum_{j=1}^{M_2}\bar{a}_j \cdot \mathrm{ReLU}(\bar{w}_jx + \bar{b}_j),$$

and defining $f_{\diamond,\iota}^{(\ell)}(h), \bar{f}_{\diamond,\iota}^{(\ell)}(h) \in \mathbb{R}^S$ for $\diamond \in \{\downarrow, \uparrow\}$ by

$$f_{\downarrow,\iota}^{(\ell)}(h)_i = \log \sum_{j=1}^S \psi_\iota^{(\ell)}(i,j) \exp(h_j), \quad \bar{f}_{\downarrow,\iota}^{(\ell)}(h)_i = \log_{\delta\star} \sum_{j=1}^S \psi_\iota^{(\ell)}(i,j) \exp_{\delta\star}(h_j), \quad \forall i \in [S],$$

$$f_{\uparrow,\iota}^{(\ell)}(h)_i = \log \sum_{j=1}^S \psi_\iota^{(\ell)}(j,i) \exp(h_j), \quad \bar{f}_{\uparrow,\iota}^{(\ell)}(h)_i = \log_{\delta\star} \sum_{j=1}^S \psi_\iota^{(\ell)}(j,i) \exp_{\delta\star}(h_j), \quad \forall i \in [S],$$

we have

$$\sup_{\max_i h_i = 0} \|f_{\diamond,\iota}^{(\ell)}(h) - \bar{f}_{\diamond,\iota}^{(\ell)}(h)\|_\infty \le \delta/(d 18^L S). \tag{30}$$

By Eq. (30) and Lemma 21, taking $b_{\uparrow,v}^{(L)} \in \mathbb{R}^S$ to be as defined in Eq. (MP-DNS) and $\bar{b}_{\uparrow,v}^{(L)} \in \mathbb{R}^S$ to be defined in Eq. (A-MP-DNS) with $\{\bar{f}_{\diamond,\iota}^{(\ell)}\}_{\ell \in [L], \iota \in [m^{(\ell)}]}$ as defined above, we have

$$\|b_{\uparrow,v}^{(L)} - \bar{b}_{\uparrow,v}^{(L)}\|_\infty \le [\delta/(d 18^L S)] \times 18^L d = \delta/S,$$

which gives

$$\sup_{z \in \mathbb{R}^d} \|m(z) - m_{\mathrm{NN}}(z)\|_\infty \le S \cdot \|b_{\uparrow,v}^{(L)} - \bar{b}_{\uparrow,v}^{(L)}\|_\infty \le \delta.$$

As a consequence, we just need to show that the approximate version of message passing algorithm as in Eq. (A-MP-DNS) could be cast as a neural network.

Indeed, by Lemma 14, there exist two-hidden-layer neural networks (for $\diamond \in \{\downarrow, \uparrow\}$, $\ell \in [L]$, and $\iota \in [m^{(\ell)}]$)

$$\mathrm{NN}_{W_{1,\diamond,\iota}^{(\ell)}, W_{2,\diamond,\iota}^{(\ell)}, W_{3,\diamond,\iota}^{(\ell)}}(h) = W_{1,\diamond,\iota}^{(\ell)} \cdot \mathrm{ReLU}(W_{2,\diamond,\iota}^{(\ell)} \cdot \mathrm{ReLU}(W_{3,\diamond,\iota}^{(\ell)} \cdot [h; 1])),$$

with $W_{1,\diamond,\iota}^{(\ell)} \in \mathbb{R}^{S \times SM_1}$, $W_{2,\diamond,\iota}^{(\ell)} \in \mathbb{R}^{SM_1 \times (SM_2+1)}$, $W_{3,\diamond,\iota}^{(\ell)} \in \mathbb{R}^{(SM_2+1) \times (S+1)}$, and

$$\|W_{1,\diamond,\iota}^{(\ell)}\|_{\max} \le 2SK, \quad \|W_{2,\diamond,\iota}^{(\ell)}\|_{\max} \le \mathrm{Poly}(SKM_1M_2), \quad \|W_{3,\diamond,\iota}^{(\ell)}\|_{\max} \le \log(4S^2 d 18^L K^2/\delta).$$

such that

$$\mathrm{NN}_{W_{1,\diamond,\iota}^{(\ell)}, W_{2,\diamond,\iota}^{(\ell)}, W_{3,\diamond,\iota}^{(\ell)}}(h) = \bar{f}_{\diamond,\iota}^{(\ell)}(h), \quad \forall h \in \mathbb{R}^S \text{ such that } \max_j h_j = 0.$$

This proves that the approximate version of message passing as in Eq. (A-MP-DNS) coincides with the U-Net as in Eq. (UNet) with proper choice of dimension

$$D \ge \max_{\ell \in [L]} \{SM_1^{(\ell)}, SM_2^{(\ell)} + 1\} = S \times \left( \lceil 2S^2 K^2 d 18^L/\delta \rceil + 1 \right) + 1$$

and norm of the weights. This finishes the proof of Theorem 6. $\qquad \square$

## G.1 AUXILLARY LEMMAS

**Lemma 21** (Error propagation of the approximate version of message passing in denoising). *Assume we have functions $\{f_{\downarrow,\iota}^{(\ell)}, f_{\uparrow,\iota}^{(\ell)}\}$ and $\{\bar{f}_{\downarrow,\iota}^{(\ell)}, \bar{f}_{\uparrow,\iota}^{(\ell)}\}$ such that*

$$\|f_{\diamond,\iota}^{(\ell)}(h) - \bar{f}_{\diamond,\iota}^{(\ell)}(h)\|_\infty \le \delta, \quad \forall h \in \mathbb{R}^S \text{ such that } \max_{j \in [S]} h_j = 0, \ \diamond \in \{\downarrow, \uparrow\}. \tag{31}$$

*Furthermore, consider the following approximate version of message passing algorithm with initialization $h_{\downarrow,v}^{(L)} = (-(x - z_v)^2/2)_{x \in [S]} \in \mathbb{R}^S$ for $v \in \mathcal{V}^{(L)}$, defined as below*

$$\bar{q}_{\downarrow,v}^{(\ell)} = \bar{f}_{\downarrow,\iota(v)}^{(\ell)}(\mathrm{normalize}(\bar{h}_{\downarrow,v}^{(\ell)})) \in \mathbb{R}^S, \qquad\qquad \ell \in [L], \quad v \in \mathcal{V}^{(\ell)},$$

$$\bar{h}_{\downarrow,v}^{(\ell-1)} = \sum_{v' \in \mathcal{C}(v)} \bar{q}_{\downarrow,v'}^{(\ell)} \in \mathbb{R}^S, \qquad\qquad\qquad \ell \in [L], \quad v \in \mathcal{V}^{(\ell-1)},$$

$$\bar{u}_{\uparrow,v}^{(\ell)} = \bar{b}_{\uparrow,\mathrm{pa}(v)}^{(\ell-1)} \in \mathbb{R}^S, \quad (\text{with } \bar{b}_{\uparrow,\mathrm{r}}^{(0)} = \bar{h}_{\downarrow,\mathrm{r}}^{(0)}) \qquad \ell \in [L], \quad v \in \mathcal{V}^{(\ell)},$$

$$\bar{b}_{\uparrow,v}^{(\ell)} = \bar{f}_{\uparrow,\iota(v)}^{(\ell)}(\mathrm{normalize}(\bar{u}_{\uparrow,v}^{(\ell)} - \bar{q}_{\downarrow,v}^{(\ell)})) + \bar{h}_{\downarrow,v}^{(\ell)} \in \mathbb{R}^S, \quad \ell \in [L], \quad v \in \mathcal{V}^{(\ell)},$$

$$\bar{m}_{\mathrm{MP}}(z)_v = \sum_{s \in [S]} s \cdot \mathrm{softmax}(\bar{b}_{\uparrow,v}^{(L)})_s, \qquad\qquad\qquad v \in \mathcal{V}^{(L)}.$$
$$\tag{A-MP-DNS}$$

*Taking $b_{\uparrow,v}^{(L)} \in \mathbb{R}^S$ to be as defined in Eq. (MP-DNS) and $\bar{b}_{\uparrow,v}^{(L)} \in \mathbb{R}^S$ to be as defined in Eq. (A-MP-DNS), we have*

$$\max_{v \in \mathcal{V}^{(L)}} \|b_{\uparrow,v}^{(L)} - \bar{b}_{\uparrow,v}^{(L)}\|_\infty \le \delta \times 18^L \cdot d.$$

*Proof of Lemma 21.*

**Step 1. Downward induction.** In the first step, we aim to show that for any $\ell \in [L-1]$ we have

$$\|\bar{h}_{\downarrow,v}^{(\ell)} - h_{\downarrow,v}^{(\ell)}\|_\infty \leq m^{(\ell+1)} \prod_{k=\ell+2}^{L}(2m^{(k)}+1)\delta, \quad \forall v \in \mathcal{V}^{(\ell)}. \tag{32}$$

To prove the formula for $\ell = L-1$, since $\mathrm{normalize}(h_{\downarrow,v}^{(L)}) = \mathrm{normalize}(\bar{h}_{\downarrow,v}^{(L)})$, by Eq. (31), we get

$$\|\bar{q}_{\downarrow,v}^{(L)} - q_{\downarrow,v}^{(L)}\|_\infty \leq \delta, \quad \forall v \in \mathcal{V}^{(L)}.$$

Hence we get

$$\|h_{\downarrow,v}^{(L-1)} - \bar{h}_{\downarrow,v}^{(L-1)}\|_\infty = \left\| \sum_{v' \in \mathcal{C}(v)} (\bar{q}_{\downarrow,v'}^{(L)} - q_{\downarrow,v'}^{(L)}) \right\|_\infty \leq m^{(L)}\delta, \quad \forall v \in \mathcal{V}^{(L-1)}.$$

This proves the formula (32) for $\ell = L-1$.

Assuming that (32) holds at the layer $\ell$, by the update formula, we have

$$\|\bar{q}_{\downarrow,v}^{(\ell)} - q_{\downarrow,v}^{(\ell)}\|_\infty = \|\bar{f}_{\downarrow,\iota(v)}^{(\ell)}(\mathrm{normalize}(\bar{h}_{\downarrow,v}^{(\ell)})) - f_{\downarrow,\iota(v)}^{(\ell)}(\mathrm{normalize}(h_{\downarrow,v}^{(\ell)}))\|_\infty$$

$$\leq \|\bar{f}_{\downarrow,\iota(v)}^{(\ell)}(\mathrm{normalize}(\bar{h}_{\downarrow,v}^{(\ell)})) - f_{\downarrow,\iota(v)}^{(\ell)}(\mathrm{normalize}(\bar{h}_{\downarrow,v}^{(\ell)}))\|_\infty + \|f_{\downarrow,\iota(v)}^{(\ell)}(\mathrm{normalize}(h_{\downarrow,v}^{(\ell)})) - f_{\downarrow,\iota(v)}^{(\ell)}(\mathrm{normalize}(\bar{h}_{\downarrow,v}^{(\ell)}))\|_\infty$$

$$\leq \delta + 2m^{(\ell+1)} \prod_{k=\ell+2}^{L}(2m^{(k)}+1)\delta \leq \prod_{k=\ell+1}^{L}(2m^{(k)}+1)\delta,$$

where the middle inequality is by the assumption of $f_{\downarrow,\iota}^{(\ell)}$ and by Lemma 6 and Lemma 7. Hence we get

$$\|\bar{h}_{\downarrow,v}^{(\ell-1)} - h_{\downarrow,v}^{(\ell-1)}\|_\infty \leq m^{(\ell)} \prod_{k=\ell+1}^{L}(2m^{(k)}+1)\delta, \quad \forall v \in \mathcal{V}^{(\ell)}.$$

This proves Eq. (32) by the induction argument.

**Step 2. Upward induction.** The downward induction argument proves that, for $\Gamma = \prod_{k=1}^{L}(2m^{(k)}+1)$, we have

$$\|\bar{q}_{\downarrow,v}^{(\ell)} - q_{\downarrow,v}^{(\ell)}\|_\infty, \|\bar{h}_{\downarrow,v}^{(\ell)} - h_{\downarrow,v}^{(\ell)}\|_\infty \leq \Gamma\delta, \quad \forall \ell = 0, 1, \ldots, L, \ \forall v \in \mathcal{V}^{(\ell)}.$$

In this step, we aim to show that for any $\ell = 0, 1, \ldots, L$, we have

$$\|\bar{b}_{\uparrow,v}^{(\ell)} - b_{\uparrow,v}^{(\ell)}\|_\infty \leq 6^\ell \cdot \Gamma \cdot \delta, \quad \forall v \in \mathcal{V}^{(\ell)}. \tag{33}$$

To prove this formula for $\ell = 0$, note that $b_{\uparrow,\mathrm{r}}^{(0)} = h_{\downarrow,\mathrm{r}}^{(0)}$ and $\bar{b}_{\uparrow,\mathrm{r}}^{(0)} = \bar{h}_{\downarrow,\mathrm{r}}^{(0)}$, we have

$$\|\bar{b}_{\uparrow,\mathrm{r}}^{(0)} - b_{\uparrow,\mathrm{r}}^{(0)}\|_\infty = \|\bar{h}_{\downarrow,\mathrm{r}}^{(0)} - h_{\downarrow,\mathrm{r}}^{(0)}\|_\infty \leq \Gamma \cdot \delta.$$

This proves the formula (33) for $\ell = 0$.

Assuming that (33) holds at layer $\ell - 1$, by the update formula, we have

$$\|b_{\uparrow,v}^{(\ell)} - \bar{b}_{\uparrow,v}^{(\ell)}\|_\infty$$

$$\leq \|f_{\uparrow,\iota(v)}^{(\ell)}(\mathrm{normalize}(b_{\uparrow,\mathrm{pa}(v)}^{(\ell-1)} - q_{\downarrow,v}^{(\ell)})) - \bar{f}_{\uparrow,\iota(v)}^{(\ell)}(\mathrm{normalize}(\bar{b}_{\uparrow,\mathrm{pa}(v)}^{(\ell-1)} - \bar{q}_{\downarrow,v}^{(\ell)}))\|_\infty + \|h_{\downarrow,v}^{(\ell)} - \bar{h}_{\downarrow,v}^{(\ell)}\|_\infty$$

$$\leq \|f_{\uparrow,\iota(v)}^{(\ell)}(\mathrm{normalize}(b_{\uparrow,\mathrm{pa}(v)}^{(\ell-1)} - q_{\downarrow,v}^{(\ell)})) - f_{\uparrow,\iota(v)}^{(\ell)}(\mathrm{normalize}(\bar{b}_{\uparrow,\mathrm{pa}(v)}^{(\ell-1)} - \bar{q}_{\downarrow,v}^{(\ell)}))\|_\infty$$

$$\quad + \|f_{\uparrow,\iota(v)}^{(\ell)}(\mathrm{normalize}(\bar{b}_{\uparrow,\mathrm{pa}(v)}^{(\ell-1)} - \bar{q}_{\downarrow,v}^{(\ell)})) - \bar{f}_{\uparrow,\iota(v)}^{(\ell)}(\mathrm{normalize}(\bar{b}_{\uparrow,\mathrm{pa}(v)}^{(\ell-1)} - \bar{q}_{\downarrow,v}^{(\ell)}))\|_\infty + \|h_{\downarrow,v}^{(\ell)} - \bar{h}_{\downarrow,v}^{(\ell)}\|_\infty$$

$$\leq \|\mathrm{normalize}(b_{\uparrow,\mathrm{pa}(v)}^{(\ell-1)} - q_{\downarrow,v}^{(\ell)}) - \mathrm{normalize}(\bar{b}_{\uparrow,\mathrm{pa}(v)}^{(\ell-1)} - \bar{q}_{\downarrow,v}^{(\ell)})\|_\infty + \delta + \Gamma \cdot \delta$$

$$\leq 4 \cdot 6^{\ell-1} \cdot \Gamma \cdot \delta + \delta + \Gamma \cdot \delta \leq 6^\ell \cdot \Gamma \cdot \delta.$$

This proves Eq. (33) by the induction argument. This proves the Lemma 21 by observing that $\Gamma \leq 3^L \prod_{\ell=1}^{L} m^{(\ell)} = 3^L \cdot d$. $\qquad\qquad\square$

# H  PROOF OF THEOREM 2

*Proof of Theorem 2.* By Lemma 22, we have the error decomposition

$$\mathsf{D}_2^2(\boldsymbol{m}_{\mathrm{NN}}^{\widehat{\boldsymbol{W}}}, \boldsymbol{m}) \leq \inf_{\boldsymbol{W} \in \mathcal{W}} \mathsf{D}_2^2(\boldsymbol{m}_{\mathrm{NN}}^{\boldsymbol{W}}, \boldsymbol{m}) + 2 \cdot \sup_{\boldsymbol{W} \in \mathcal{W}} \left| \widehat{\mathsf{R}}(\boldsymbol{m}_{\mathrm{NN}}^{\boldsymbol{W}}) - \mathsf{R}(\boldsymbol{m}_{\mathrm{NN}}^{\boldsymbol{W}}) \right|.$$

To control the first term (the approximation error), by Theorem 6, there exists $\boldsymbol{W} \in \mathcal{W}_{d,\underline{m},L,S,D,B}$ as in Eq. (18) with norm bound $B = \mathrm{Poly}(d, S, K, 18^L, D)$, such that defining $\boldsymbol{m}_{\mathrm{NN}}^{\boldsymbol{W}}$ as in Eq. (UNet), we have

$$\sup_{\boldsymbol{z} \in \mathbb{R}^d} \|\boldsymbol{m}(\boldsymbol{z}) - \boldsymbol{m}_{\mathrm{NN}}(\boldsymbol{z})\|_\infty \leq C \cdot \frac{S^3 K^2 d \cdot 18^L}{D}.$$

Therefore, we have

$$\inf_{\boldsymbol{W} \in \mathcal{W}} \mathsf{D}_2^2(\boldsymbol{m}_{\mathrm{NN}}^{\boldsymbol{W}}, \boldsymbol{m}) \leq \sup_{\boldsymbol{z}} \|\boldsymbol{m}(\boldsymbol{z}) - \boldsymbol{m}_{\mathrm{NN}}(\boldsymbol{z})\|_\infty^2 \leq C \cdot \frac{S^6 K^4 d^2 \cdot 18^{2L}}{D^2}.$$

To control the second term (the generalization error), by Proposition 8, with probability at least $1 - \eta$, we have

$$\sup_{\boldsymbol{W} \in \mathcal{W}_{d,\underline{m},L,S,D,B}} \left| \widehat{\mathsf{R}}(\boldsymbol{m}_{\mathrm{NN}}^{\boldsymbol{W}}) - \mathsf{R}(\boldsymbol{m}_{\mathrm{NN}}^{\boldsymbol{W}}) \right| \leq C \cdot S^2 \cdot \sqrt{\frac{LD(D + 2S + 1)\|\underline{m}\|_1 \log(d\|\underline{m}\|_1 DSB \cdot 18^L) + \log(1/\eta)}{n}}.$$

Combining the above two equations proves Theorem 2. $\qquad\square$

## H.1  ERROR DECOMPOSITION

**Lemma 22.** *Consider the setting of Theorem 2. We have decomposition*

$$\mathsf{D}_2^2(\boldsymbol{m}_{\mathrm{NN}}^{\widehat{\boldsymbol{W}}}, \boldsymbol{m}) \leq \inf_{\boldsymbol{W} \in \mathcal{W}} \mathsf{D}_2^2(\boldsymbol{m}_{\mathrm{NN}}^{\boldsymbol{W}}, \boldsymbol{m}) + 2 \cdot \sup_{\boldsymbol{W} \in \mathcal{W}} \left| \widehat{\mathsf{R}}(\boldsymbol{m}_{\mathrm{NN}}^{\boldsymbol{W}}) - \mathsf{R}(\boldsymbol{m}_{\mathrm{NN}}^{\boldsymbol{W}}) \right|.$$

*Proof of Lemma 22.* We have that for any conditional expectation $\boldsymbol{m}_1(\boldsymbol{z})$, there is decomposition

$$\mathsf{D}_2^2(\boldsymbol{m}_1, \boldsymbol{m}) = \mathbb{E}_{(\boldsymbol{x}, \boldsymbol{z}) \sim \mu_\star} \left[ d^{-1} \|\boldsymbol{m}_1(\boldsymbol{z}) - \boldsymbol{m}(\boldsymbol{z})\|_2^2 \right]$$

$$= \mathbb{E}_{(\boldsymbol{x}, \boldsymbol{z}) \sim \mu_\star} \left[ d^{-1} \|\boldsymbol{m}_1(\boldsymbol{z}) - \boldsymbol{x}\|_2^2 \right] - \mathbb{E}_{(\boldsymbol{x}, \boldsymbol{z}) \sim \mu_\star} \left[ d^{-1} \|\boldsymbol{m}_1(\boldsymbol{z}) - \boldsymbol{x}\|_2^2 \right] = \mathsf{R}(\boldsymbol{m}_1) - \mathsf{R}(\boldsymbol{m}).$$

Define

$$\boldsymbol{W}_\star = \arg \min_{\boldsymbol{W} \in \mathcal{W}} \mathsf{R}(\boldsymbol{m}_{\mathrm{NN}}^{\boldsymbol{W}}) = \arg \min_{\boldsymbol{W} \in \mathcal{W}} \mathsf{D}_2^2(\boldsymbol{m}_{\mathrm{NN}}^{\boldsymbol{W}}, \boldsymbol{m}).$$

Then we have

$$\mathsf{D}_2^2(\boldsymbol{m}_{\mathrm{NN}}^{\widehat{\boldsymbol{W}}}, \boldsymbol{m}) = \mathsf{R}(\boldsymbol{m}_{\mathrm{NN}}^{\widehat{\boldsymbol{W}}}) - \mathsf{R}(\boldsymbol{m})$$

$$= \mathsf{R}(\boldsymbol{m}_{\mathrm{NN}}^{\widehat{\boldsymbol{W}}}) - \widehat{\mathsf{R}}(\boldsymbol{m}_{\mathrm{NN}}^{\widehat{\boldsymbol{W}}}) + \widehat{\mathsf{R}}(\boldsymbol{m}_{\mathrm{NN}}^{\widehat{\boldsymbol{W}}}) - \widehat{\mathsf{R}}(\boldsymbol{m}_{\mathrm{NN}}^{\boldsymbol{W}_\star}) + \widehat{\mathsf{R}}(\boldsymbol{m}_{\mathrm{NN}}^{\boldsymbol{W}_\star}) - \mathsf{R}(\boldsymbol{m}_{\mathrm{NN}}^{\boldsymbol{W}_\star}) + \mathsf{R}(\boldsymbol{m}_{\mathrm{NN}}^{\boldsymbol{W}_\star}) - \mathsf{R}(\boldsymbol{m})$$

$$\leq 2 \cdot \sup_{\boldsymbol{W} \in \mathcal{W}} \left| \mathsf{R}(\boldsymbol{m}_{\mathrm{NN}}^{\boldsymbol{W}}) - \widehat{\mathsf{R}}(\boldsymbol{m}_{\mathrm{NN}}^{\boldsymbol{W}}) \right| + \mathsf{D}_2^2(\boldsymbol{m}_{\mathrm{NN}}^{\boldsymbol{W}_\star}, \boldsymbol{m})$$

This proves Lemma 22. $\qquad\square$

## H.2  RESULTS ON GENERALIZATION

**Proposition 8** (Generalization error of the denoising problem). *Let $\mathcal{W}_{d,\underline{m},L,S,D,B}$ be the set defined as in Eq. (9). Then, with probability at least $1 - \eta$, we have*

$$\sup_{\boldsymbol{W} \in \mathcal{W}_{d,\underline{m},L,S,D,B}} \left| \widehat{\mathsf{R}}(\mu_{\mathrm{NN}}^{\boldsymbol{W}}) - \mathsf{R}(\mu_{\mathrm{NN}}^{\boldsymbol{W}}) \right| \leq C \cdot S^2 \cdot \sqrt{\frac{LD(D + 2S + 1)\|\underline{m}\|_1 \log(d\|\underline{m}\|_1 DSB \cdot 18^L) + \log(1/\eta)}{n}}.$$

*Proof of Proposition 8.* In Lemma 3, we can take $z = (\boldsymbol{z}, \boldsymbol{x})$, $w = \boldsymbol{W}$, $\Theta = \mathcal{W}_{d,\underline{m},L,S,D,B}$, $\rho(w, w') = \|\boldsymbol{W} - \boldsymbol{W}'\|$, and $f(z_i; w) = \|\boldsymbol{x} - \boldsymbol{m}_{\mathrm{NN}}^{\boldsymbol{W}}(\boldsymbol{z})\|_2^2$. Therefore, to show Proposition 8, we just need to apply Lemma 3 by checking (a), (b), (c).

**Check (a).** We note that the index set $\Theta := \mathcal{W}_{d,\underline{m},L,S,D,B}$ equipped with $\rho(w, w') := \|\boldsymbol{W} - \boldsymbol{W}'\|$ has diameter $B_{\mathrm{p}} := 2B$. Further note that $\mathcal{W}_{d,\underline{m},L,S,D,B}$ has a dimension bounded by $d_{\mathrm{p}} := 2D(D + 2S + 1)\|\underline{m}\|_1$. According to Example 5.8 of (Wainwright, 2019), it holds that $\log N(\Delta; \mathcal{W}_{d,\underline{m},L,S,D,B}, \|\cdot\|) \le d_{\mathrm{p}} \cdot \log(1 + 2r/\Delta)$ for any $0 < \Delta \le 2r$. This verifies (a).

**Check (b).** Since $f(z_i; w) = d^{-1}\|\boldsymbol{x} - \boldsymbol{m}_{\mathrm{NN}}^{\boldsymbol{W}}(\boldsymbol{z})\|_2^2$ is $S^2$-bounded. As a consequence, $f(z, w) - \mathbb{E}_z[f(z, w)]$ is a sub-Gaussian random variable with the sub-Gaussian parameter to be $C \cdot S^2$.

**Check (c).** Lemma 25 implies that

$$|f(z; w_1) - f(z; w_2)| \le L_{\mathrm{p}} \cdot \|\boldsymbol{W}_1 - \boldsymbol{W}_2\|, \quad L_{\mathrm{p}} := 12\|\underline{m}\|_1 18^L B^{6L} \cdot d \cdot S^3 \Big( \sum_{v \in \mathcal{V}^{(L)}} (S + |z_v|) \Big) \cdot (S + D).$$

Since $\boldsymbol{z} \overset{d}{=} \boldsymbol{x} + \boldsymbol{g}$ where $(\boldsymbol{x}, \boldsymbol{g}) \sim \mu_\star \times \mathcal{N}(\boldsymbol{0}, \mathbf{I}_d)$, and $\|\boldsymbol{x}\|_1 \le Sd$, $\|\boldsymbol{g}\|_1$ is $Cd$-sub-Gaussian. Hence $\|\boldsymbol{z}\|_1$ is $CSd$-sub-Gausssian, and hence $f(z; w_1) - f(z; w_2)$ is $\sigma'\rho(w_1, w_2)$ sub-Gaussian with

$$\sigma' = C\|\underline{m}\|_1 18^L B^{6L} \cdot d^2 \cdot S^4 \cdot (S + D).$$

Therefore, we apply Lemma 3 to conclude the proof of Proposition 8. $\qquad\square$

### H.3 AUXILLARY LEMMAS

**Lemma 23** (Norm bound in the chain rule in denoising settings)**.** *Consider the U-Net as in Eq. (UNet) with modified input* $\mathsf{h}_{\downarrow,v}^{(L)} = (-x^2/2 + xz_v)_{x\in[S]} \in \mathbb{R}^S$ *(Since we will immediately normalize the input, this input is effectively the same as the input* $\mathsf{h}_{\downarrow,v}^{(L)} = (-(x - z_v)^2/2)_{x\in[S]} \in \mathbb{R}^S$*). Assume that* $\|\boldsymbol{W}\| \le B$*. Then for any* $\ell$*,* $v$*,* $\iota$*, and* $\star \in [3]$*, we have*

$$\|\mathsf{h}_{\downarrow,v}^{(L)}\|_2 \le S^3 + S^2|z_v|,$$

$$\|\mathsf{q}_{\downarrow,v}^{(\ell)}\|_2 \le B^3 \cdot (2 \cdot \|\mathsf{h}_{\downarrow,v}^{(\ell)}\|_2 + 1),$$

$$\|\mathsf{h}_{\downarrow,v}^{(\ell-1)}\|_2 \le m^{(\ell)} \cdot \max_{v'\in\mathcal{C}(v)} \|\mathsf{q}_{\downarrow,v'}^{(\ell)}\|_2,$$

$$\|\mathsf{b}_{\uparrow,\mathrm{r}}^{(0)}\|_2 = \|\mathsf{h}_{\downarrow,\mathrm{r}}^{(0)}\|_2,$$

$$\|\mathsf{b}_{\uparrow,v}^{(\ell)}\|_2 \le B^3 \cdot (2\|\mathsf{b}_{\uparrow,\mathrm{pa}(v)}^{(\ell-1)}\|_2 + 2\|\mathsf{q}_{\downarrow,v}^{(\ell)}\|_2 + 1) + \|\mathsf{h}_{\downarrow,v}^{(\ell)}\|_2,$$

$$\max_{i\in[S]} \|\nabla_{W_{\star,\downarrow,\iota}^{(k)}} \mathsf{q}_{\downarrow,v,i}^{(\ell)}\|_{\mathrm{op}} \le 2B^3 \cdot \max_{i\in[S]} \|\nabla_{W_{\star,\downarrow,\iota}^{(k)}} \mathsf{h}_{\downarrow,v,i}^{(\ell)}\|_{\mathrm{op}}, \qquad \forall k \ge \ell+1,$$

$$\max_{i\in[S]} \|\nabla_{W_{\star,\downarrow,\iota}^{(\ell)}} \mathsf{q}_{\downarrow,v,i}^{(\ell)}\|_{\mathrm{op}} \le 2B^2 \cdot \|\mathsf{h}_{\downarrow,v}^{(\ell)}\|_2,$$

$$\max_{i\in[S]} \|\nabla_{W_{\star,\downarrow,\iota}^{(k)}} \mathsf{h}_{\downarrow,v,i}^{(\ell-1)}\|_{\mathrm{op}} \le m^{(\ell)} \cdot \max_{v'\in\mathcal{C}(v)} \max_{i\in[S]} \|\nabla_{W_{\star,\downarrow,\iota}^{(k)}} \mathsf{q}_{\downarrow,v',i}^{(\ell)}\|_{\mathrm{op}}, \qquad \forall k \ge \ell,$$

$$\max_{i\in[S]} \|\nabla_{W_{\star,\uparrow,\iota}^{(k)}} \mathsf{b}_{\uparrow,v,i}^{(\ell)}\|_{\mathrm{op}} \le 2B^3 \cdot \max_{i\in[S]} \|\nabla_{W_{\star,\uparrow,\iota}^{(k)}} \mathsf{b}_{\uparrow,\mathrm{pa}(v),i}^{(\ell-1)}\|_{\mathrm{op}}, \qquad \forall k \ge \ell+1,$$

$$\max_{i\in[S]} \|\nabla_{W_{\star,\uparrow,\iota}^{(\ell)}} \mathsf{b}_{\uparrow,v,i}^{(\ell)}\|_{\mathrm{op}} \le 2B^2 \cdot (\|\mathsf{b}_{\uparrow,\mathrm{pa}(v)}^{(\ell-1)}\|_2 + \|\mathsf{q}_{\downarrow,v}^{(\ell)}\|_2),$$

$$\max_{i\in[S]} \|\nabla_{W_{\star,\downarrow,\iota}^{(k)}} \mathsf{b}_{\uparrow,v,i}^{(\ell)}\|_{\mathrm{op}} \le 2B^3 \cdot \Big( \max_{i\in[S]} \|\nabla_{W_{\star,\downarrow,\iota}^{(k)}} \mathsf{b}_{\uparrow,\mathrm{pa}(v),i}^{(\ell-1)}\|_{\mathrm{op}}$$

$$+ \max_{i\in[S]} \|\nabla_{W_{\star,\downarrow,\iota}^{(k)}} \mathsf{q}_{\downarrow,v,i}^{(\ell)}\|_{\mathrm{op}} \Big) + \max_{i\in[S]} \|\nabla_{W_{\star,\downarrow,\iota}^{(k)}} \mathsf{h}_{\downarrow,v,i}^{(\ell)}\|_{\mathrm{op}}, \quad \forall k \in [L],$$

$$\|\nabla_{W_{\star,\diamond,\iota}^{(k)}} \boldsymbol{m}_{\mathrm{NN}}^{\boldsymbol{W}}(\boldsymbol{z})_v\|_{\mathrm{op}} \le S \cdot \max_{i\in[S]} \|\nabla_{W_{\star,\diamond,\iota}^{(k)}} \mathsf{b}_{\uparrow,v,i}^{(L)}\|_{\mathrm{op}}, \qquad \forall k \in [L], \diamond \in \{\downarrow, \uparrow\}.$$

*Proof of Lemma 23.* The proof of the lemma uses the chain rule, the 1-Lipschitzness of ReLU, the 2-Lipschitzness of normalize, and the 1-Lipschitzness of softmax. $\qquad\square$

**Lemma 24.** *Consider the U-Net as in Eq. (UNet). Assume that $\|\boldsymbol{W}\| \leq B$. Then for any $v \in \mathcal{V}^{(L)}$, we have*

$$\max_{\diamond \in \{\downarrow, \uparrow\}} \max_{\boldsymbol{z} \in \mathbb{R}^d} \max_{\star \in [3]} \max_{\ell \in [L]} \max_{\iota \in [m^{(\ell)}]} \|\nabla_{W_{\star,\diamond,\iota}^{(\ell)}} \boldsymbol{m}_{\mathrm{NN}}^{\boldsymbol{W}}(\boldsymbol{z})_v\|_{\mathrm{op}} \leq 18^L B^{6L} \cdot d \cdot S^3 (S + |z_v|).$$

*Proof of Lemma 24.* This lemma is implied by Lemma 23 and an induction argument. □

**Lemma 25.** *Consider the ConvNet as in Eq. (UNet). Assume that $\|\boldsymbol{W}\| \leq B$. Then for any $v \in \mathcal{V}^{(L)}$, we have*

$$\max_{\boldsymbol{z}} \left| \boldsymbol{m}_{\mathrm{NN}}^{\boldsymbol{W}}(\boldsymbol{z})_v - \boldsymbol{m}_{\mathrm{NN}}^{\overline{\boldsymbol{W}}}(\boldsymbol{z})_v \right| \leq 6\|\underline{m}\|_1 18^L B^{6L} \cdot d \cdot S^3 (S + |z_v|) \cdot (S + D) \cdot \|\boldsymbol{W} - \overline{\boldsymbol{W}}\|.$$

*Therefore, we have*

$$\max_{\boldsymbol{z}} d^{-1} \left| \|\boldsymbol{x} - \boldsymbol{m}_{\mathrm{NN}}^{\boldsymbol{W}}(\boldsymbol{z})\|_2^2 - \|\boldsymbol{x} - \boldsymbol{m}_{\mathrm{NN}}^{\overline{\boldsymbol{W}}}(\boldsymbol{z})\|_2^2 \right| \leq 12\|\underline{m}\|_1 18^L B^{6L} \cdot d \cdot S^3 \left( \sum_{v \in \mathcal{V}^{(L)}} (S + |z_v|) \right) \cdot (S + D) \cdot \|\boldsymbol{W} - \overline{\boldsymbol{W}}\|.$$

*Proof of Lemma 25.* The first inequality is by the fact that

$$\max_{\boldsymbol{z}} \left| \boldsymbol{m}_{\mathrm{NN}}^{\boldsymbol{W}}(\boldsymbol{z})_v - \boldsymbol{m}_{\mathrm{NN}}^{\overline{\boldsymbol{W}}}(\boldsymbol{z})_v \right|$$
$$\leq \sum_{\diamond \in \{\downarrow, \uparrow\}} \sum_{\star \in [3]} \sum_{\ell \in [L]} \sum_{\iota \in [m^{(\ell)}]} \min\{\mathrm{nrow}(W_{\star,\diamond,\iota}^{(\ell)}), \mathrm{ncol}(W_{\star,\diamond,\iota}^{(\ell)})\} \|\nabla_{W_{\star,\diamond,\iota}^{(\ell)}} \boldsymbol{m}_{\mathrm{NN}}^{\tilde{\boldsymbol{W}}}(\boldsymbol{z})_v\|_{\mathrm{op}} \|W_{\star,\diamond,\iota}^{(\ell)} - \overline{W}_{\star,\diamond,\iota}^{(\ell)}\|_{\mathrm{op}},$$

where we have used the inequality that $\mathrm{trace}(A^{\mathsf{T}} B) \leq \{\mathrm{nrow}(A), \mathrm{ncol}(A)\} \|A\|_{\mathrm{op}} \|B\|_{\mathrm{op}}$.

To prove the second inequality, we have

$$\max_{\boldsymbol{z}} \left| \|\boldsymbol{x} - \boldsymbol{m}_{\mathrm{NN}}^{\boldsymbol{W}}(\boldsymbol{z})\|_2^2 - \|\boldsymbol{x} - \boldsymbol{m}_{\mathrm{NN}}^{\overline{\boldsymbol{W}}}(\boldsymbol{z})\|_2^2 \right|$$
$$\leq \max_{\boldsymbol{z}} \|2\boldsymbol{x} - \boldsymbol{m}_{\mathrm{NN}}^{\boldsymbol{W}}(\boldsymbol{z}) - \boldsymbol{m}_{\mathrm{NN}}^{\overline{\boldsymbol{W}}}(\boldsymbol{z})\|_\infty \|\boldsymbol{m}_{\mathrm{NN}}^{\boldsymbol{W}}(\boldsymbol{z}) - \boldsymbol{m}_{\mathrm{NN}}^{\overline{\boldsymbol{W}}}(\boldsymbol{z})\|_1$$
$$\leq 2d \max_{\boldsymbol{z}} \|\boldsymbol{m}_{\mathrm{NN}}^{\boldsymbol{W}}(\boldsymbol{z}) - \boldsymbol{m}_{\mathrm{NN}}^{\overline{\boldsymbol{W}}}(\boldsymbol{z})\|_1.$$

This completes the proof of Lemma 25. □

