# OpenReview forum: "U-Nets as Belief Propagation: Efficient Classification, Denoising, and Diffusion in Generative Hierarchical Models"
_ICLR.cc/2025/Conference — ICLR 2025 Poster_

### Official Review · Reviewer_5xWe · 2024-11-04

**Soundness:** 3
**Presentation:** 4
**Contribution:** 3
**Rating:** 6
**Confidence:** 2

**Summary:**

This paper provides a theoretical framework for understanding U-Nets. The authors present the connection between U-Nets and belief propagation algorithms within generative hierarchical models (GHMs). They show how the components of the U-Net model naturally implement belief propagation steps, which enable efficient denoising operations. In particular, they prove how they can approximate the message parsing algorithm and, therefore, also the belief propagation algorithm, using ConvNets and U-Nets for classification and denoising tasks, respectively. Finally, based on these proofs, this paper establishes sample complexity bounds for learning ConvNet classifiers and denoising functions with U-Nets.

**Strengths:**

The paper is very well-written. It is well-organized, with a clear outline and structure that facilitates understanding. The presentation of the generative hierarchical model, followed by the classification problem with ConvNets and the Denoising Problem with U-Nets is impeccable.

The authors demonstrate strong theoretical rigor. They present detailed and thorough proofs in a way that encourages replication and verification. The paper leaves no gaps in its logical arguments and even addresses potential objections given some simplifying assumptions made by the authors.

The literature review is extensive and effectively situates the contributions of the papers within the current research, by highlighting its value of adding a theoretical foundation for U-Nets and presenting the first sample complexity results for learning a Bayes denoiser using U-Nets.

**Weaknesses:**

The paper makes simplifying assumptions regarding “convolutional layers” and the use of square loss instead of cross-entropy. While it is helpful that the authors clarify these deviations from practical implementations, referring to these layers as “convolutional” may be misleading, as they do not fully adhere to the standard convolution operations. However, it is noted that parameter sharing and local invariance properties are retained, though it remains unclear whether true convolution or cross-correlation is used.

The practical utility of the provided bounds is unclear, as their direct practical applicability for empirical settings is not fully established.

**Questions:**

Can the authors clarify whether true convolution or cross-correlation is used, given that parameter sharing and local invariance properties are retained?

What are the potential applications of the established bounds? How could the theoretical findings be directly applied to or inform improvements in state-of-the-art diffusion models?

How do the results theoretically align with empirical observations in commonly used U-Net-based models for diffusion, such as those in image generation tasks? Could the authors elaborate on any observed gaps between theory and practice?

---

> ### Author Response · Authors · 2024-11-17
>
> We thank the reviewer for their thoughtful comments. Below, we address the identified weaknesses and questions.
>
> > "The paper makes simplifying assumptions regarding “convolutional layers” and the use of square loss instead of cross-entropy. While it is helpful that the authors clarify these deviations from practical implementations, referring to these layers as “convolutional” may be misleading, as they do not fully adhere to the standard convolution operations. However, it is noted that parameter sharing and local invariance properties are retained, though it remains unclear whether true convolution or cross-correlation is used. Can the authors clarify whether true convolution or cross-correlation is used, given that parameter sharing and local invariance properties are retained? "
>
> The convolution operation used in real ConvNets differs from the one in this paper in the following way: in standard ConvNets, each image patch undergoes an inner product with a convolution filter (a linear transformation of each pixel and aggregation of pixel information), followed by a ReLU activation function (pointwise nonlinearity). In contrast, in this paper, each image pixel is first processed by a nonlinear transformation (multiple linear transformations and pointwise nonlinearities) before the pixel information is aggregated. The key distinction lies in the order of operations—“linear transformation”, “pointwise nonlinearity”, and “pixel information aggregation”, while translation invariance is preserved.
>
> After submitting the paper, we realized that multiple real convolutional layers could simulate the convolutional layer used in our paper. Specifically, each real convolution filter would need to have only one vector of non-zero weights corresponding to a single pixel, with all other pixel weights set to zero. While this pattern is not typically observed in practice, it is theoretically valid and can simulate the pixel-wise linear transformation employed in our paper. (We also notice that ​​in real U-Nets, people indeed use multiple convolutional layers at each level) We will include this clarification in the final version of the paper.
>
>
> > "The practical utility of the provided bounds is unclear, as their direct practical applicability for empirical settings is not fully established. What are the potential applications of the established bounds? How could the theoretical findings be directly applied to or inform improvements in state-of-the-art diffusion models?"
>
> The theoretical bound avoids the curse of dimensionality in neural network approximation, demonstrating the efficiency of neural networks in hierarchical models. While the bound itself may not have immediate practical applications, the intuition provided by the theoretical results could be insightful.
>
> In proving the theoretical results, we leverage the fact that each layer of the U-Net approximates a belief-propagation step in the denoising process. This suggests a hypothesis that the intermediate layers of U-Nets, after training, represent intermediate-level concepts within a hierarchical model. This hypothesis is supported by several works, including [KGMS23] and [SFW23]. Future research could explore this observation for algorithm design. For instance, the theoretical construction suggests that the middle layers may represent high-level image concepts, such as “birds” or “dogs.” These intermediate representations could potentially serve as useful features for the image classification task.
>
> [KGMS23] Zahra Kadkhodaie, Florentin Guth, Stephane Mallat, and Eero P Simoncelli. Learning multi-scale local conditional probability models of images. ICLR 2023.
>
> [SFW24] Antonio Sclocchi, Alessandro Favero, and Matthieu Wyart. A phase transition in diffusion models reveals the hierarchical nature of data. arXiv preprint arXiv:2402.16991, 2024.
>
> > "How do the results theoretically align with empirical observations in commonly used U-Net-based models for diffusion, such as those in image generation tasks? Could the authors elaborate on any observed gaps between theory and practice?"
>
> Regarding the alignment between theory and practice: as mentioned in response to the previous question, the theory predicts that the intermediate layers of U-Nets represent intermediate-level concepts of a transformer. This hypothesis is partially supported by several works that have conducted experimental studies of U-Nets, including [KGMS23] and [SFW23].
>
> Some gaps between theory and practice remain. In most practical diffusion models, particularly those used for image generation, the models are conditional, with text input serving as guidance. Our current model does not incorporate the text input. Future work will explore joint modeling of both image and text distributions and investigate the architecture for joint belief propagation processes.

---

> > ### Comment · Reviewer_5xWe · 2024-11-18
> > **Reply to the Comment by the Authors**
> >
> > I thank the authors for their thoughtful responses to my comments and questions.
> >
> > I appreciate the clarifications regarding the convolutional layers. The discussion of the gaps between theory and practice and the acknowledgment of future directions, such as incorporating conditional models and text input, also strengthen the work's relevance and potential impact.
> >
> > Based on my review, I am confident that this paper is well-written, presents a sound contribution, and is worthy of acceptance. However, given the complexity and extensive nature of the proofs provided in the appendix, I must acknowledge that I did not carefully verify every step of the theoretical arguments. From the main text and the portions of the appendix that I reviewed, the work appears to be rigorous and sound.
> >
> > I am willing to defend the score I have assigned. However, I cannot increase it as I was not able to thoroughly verify all the steps in the extensive proofs provided in the appendix.

---

> > > ### Author Response · Authors · 2024-11-22
> > >
> > > We truly appreciate your positive feedback and thoughtful comments! We hope you might support the paper during discussions. Thank you!

---

### Official Review · Reviewer_xjiq · 2024-11-04

**Soundness:** 2
**Presentation:** 3
**Contribution:** 2
**Rating:** 6
**Confidence:** 2

**Summary:**

This paper presents a new perspective on the U-Net architecture, linking it to tree-structured models and the belief propagation denoising algorithm. The authors derive sample complexity bounds for U-Nets in learning the denoising function, employing a proof strategy similar to that used for ConvNets in classification tasks.

**Strengths:**

(1) Authors offer a novel interpretation of U-Nets by linking them to the belief propagation algorithm. It’s intriguing how this algorithm naturally gives rise to key U-Net components, such as skip connections and pooling layers.

(2) Authors conduct a thorough complexity analysis, with extensive proofs provided in the appendix.

(3) The paper is well-written, and the proof strategy in the ConvNet example aids in understanding the later sections on U-Nets.

**Weaknesses:**

(1) Although this is a theoretical paper, it lacks any form of empirical validation of its conclusions, even in a simplified format.

(2) Why is the belief propagation algorithm applied exclusively to U-Nets? For instance, ResNet’s residual connections could also approximate “long skip connections” that link all nodes, disregarding the up/down sampling operations.

(3) The title suggests that U-Nets are efficient for classification, yet the paper only analyzes classification in the context of ConvNets.

**Questions:**

See the Weakness.

---

> ### Author Response · Authors · 2024-11-17
>
> We thank the reviewer for their thoughtful comments. Below, we address the identified weaknesses and questions.
>
> > "Although this is a theoretical paper, it lacks any form of empirical validation of its conclusions, even in a simplified format."
>
> While we did not conduct numerical simulations, numerous experiments in the literature demonstrate the effectiveness of U-Nets in image-denoising tasks and hierarchical denoising models, including [KGMS23] and [SFW24]. We believe these existing experiments provide good empirical validation for our theoretical framework.
>
> We acknowledge the importance of experimental validation to make the paper more self-contained and regret that we may not be able to complete simulations within the short term. Nonetheless, we plan to include experimental results in the final version.
>
> [KGMS23] Zahra Kadkhodaie, Florentin Guth, Stephane Mallat, and Eero P Simoncelli. Learning multi-scale local conditional probability models of images. ICLR 2023.
>
> [SFW24] Antonio Sclocchi, Alessandro Favero, and Matthieu Wyart. A phase transition in diffusion models reveals the hierarchical nature of data. arXiv preprint arXiv:2402.16991, 2024.
>
> > "Why is the belief propagation algorithm applied exclusively to U-Nets? For instance, ResNet’s residual connections could also approximate “long skip connections” that link all nodes, disregarding the up/down sampling operations."
>
> We agree that ResNets (with short skip connections, assuming all intermediate layer dimensions are kept the same) and transformers could also simulate the belief propagation process and serve as effective denoising functions. In principle, we could theoretically demonstrate that ResNets and transformers can approximate belief propagation in the GHM framework, though they might require more parameters than U-Nets. That said, U-Nets are not the only architectures capable of performing well on denoising tasks.
>
> However, ResNets and transformers may involve redundant operations that could make them less efficient than U-Nets in hierarchical models, given the same number of training samples. Intuitively, U-Nets represent a minimal structure that effectively parameterizes the belief propagation process, making them particularly well-suited for such tasks.
>
> > "The title suggests that U-Nets are efficient for classification, yet the paper only analyzes classification in the context of ConvNets."
>
> We apologize for the confusion caused by the title. Our intention was not to suggest that U-Nets are efficient for classification. Instead, we aimed to demonstrate that ConvNets are efficient for classification tasks, while U-Nets are efficient for denoising and diffusion tasks.

---

> > ### Author Response · Authors · 2024-11-22
> >
> > We wanted to follow up on our previous response to your comments. We hope our clarifications address your concerns, and we’d be happy to answer your further questions.

---

> ### Comment · Reviewer_xjiq · 2024-11-26
> **Thanks for the feedback**
>
> I appreciate the authors' feedback. It is understandable that the authors were unable to include toy experiments to quantify the impact of the sample complexity advantages at this stage. However, as promised, please ensure these results are included in the final version. Additionally, kindly update the text to clarify the connection between the UNet and classification to avoid confusion. Beyond these points, I believe the authors have sufficiently addressed my concerns, and I am inclined to raise the score to a 6.

---

> > ### Author Response · Authors · 2024-11-26
> >
> > We appreciate the reviewer raising the score. We will ensure that the requested experiments and the clarification are incorporated into the final version.

---

### Official Review · Reviewer_UKfv · 2024-11-04

**Soundness:** 4
**Presentation:** 3
**Contribution:** 3
**Rating:** 8
**Confidence:** 4

**Summary:**

This paper studies approximation and generalization of the classification function / score function in diffusion models. The data is assumed to satisfy a tree-structured Markov random field model with hierarchical latent variables. The authors show how respectively a (variant of) CNN and UNet architectures are tailored for this problem, due to their resemblance to the belief-propagation algorithm. They obtain polynomial rates for the approximation and generalization error in terms of the network width and number of samples, showing that these architectures together with the assumptions on the data are able to break the curse of dimensionality.

**Strengths:**

This paper studies an important problem, which is to understand what properties of the data distribution coupled with inductive biases of network architectures allow to break the curse of dimensionality when learning diffusion models. The observation that CNNs and UNet correspond to classical Bayesian inference algorithms in the case of tree-structured models is an interesting one. To the best of my knowledge, this is the first work that obtains such results for deep networks.

**Weaknesses:**

The main weakness of the paper is that the data assumptions are very restrictive. In particular, they exist simpler and more efficient classical algorithms to learn and sample from these distributions than diffusion models.

**Questions:**

- Why is the assumption that children are independent when conditioned on their parent necessary? Belief propagation would still be possible (with a computational complexity exponential in the branching factor, but this would remain manageable if it is small).
- Do the authors believe that their model captures the relevant properties of image distributions? How would they go about testing it? And if no, what in their opinion are the missing features that should be studied in future work?

---

> ### Author Response · Authors · 2024-11-17
>
> We thank the reviewer for supporting our paper. Below, we address the identified weaknesses and questions.
>
> > "The main weakness of the paper is that the data assumptions are very restrictive. … Why is the assumption that children are independent when conditioned on their parent necessary? Belief propagation would still be possible (with a computational complexity exponential in the branching factor, but this would remain manageable if it is small). "
>
> Belief propagation is indeed possible and efficient as long as the number of children per node is small. The conditional independence assumption serves as a simplification and proof of concept. We believe that even without this assumption, through a more careful analysis of approximation theory, the denoising function would still admit efficient U-Net approximation.
>
> > "In particular, they exist simpler and more efficient classical algorithms to learn and sample from these distributions than diffusion models. "
>
> We agree that simpler and more efficient classical algorithms exist for learning and sampling from such distributions. For instance, a classical approach could involve using a maximum likelihood estimator to learn the graphical model and then running belief propagation on the learned model for denoising.
>
> Our theoretical contribution demonstrates that diffusion models can also perform the same task efficiently. Conceptually, diffusion models have the advantage of relying less on the precise specification of the graphical model. Classical approaches may suffer when the model is misspecified, whereas diffusion models learn the score function directly using neural networks without explicit knowledge of the data-generating process. This versatility makes diffusion models particularly appealing for applications where the data-generating process is unknown.
>
> > Do the authors believe that their model captures the relevant properties of image distributions? How would they go about testing it? And if no, what in their opinion are the missing features that should be studied in future work?
>
> We believe the GHM captures some key properties of image distributions. There is existing work demonstrating the effectiveness of hierarchical models (not necessarily with the conditional independence assumption) in modeling image distributions, such as [KGMS23], [SFW24], and [PCTFW23].
>
> However, the GHM proposed in this paper has limitations, which could be addressed in future work: (1) The conditional independence assumption is restrictive and could be generalized to better reflect real-world data. (2) Each hidden node in the current model is assumed to be a discrete variable taking values in [S]. Extending the framework to include continuous variables would be an interesting and valuable direction.
>
> [KGMS23] Zahra Kadkhodaie, Florentin Guth, Stephane Mallat, and Eero P Simoncelli. Learning multi-scale local conditional probability models of images. ICLR 2023.
>
> [SFW24] Antonio Sclocchi, Alessandro Favero, and Matthieu Wyart. A phase transition in diffusion models reveals the hierarchical nature of data. arXiv preprint arXiv:2402.16991, 2024.
>
> [PCTFW23] Leonardo Petrini, Francesco Cagnetta, Umberto M Tomasini, Alessandro Favero, and Matthieu Wyart. How deep neural networks learn compositional data: The random hierarchy model. arXiv preprint arXiv:2307.02129, 2023.

---

> > ### Comment · Reviewer_UKfv · 2024-11-21
> >
> > I thank the authors for their response. I think it could be worth to point out that the conditional independence assumption is mostly a matter of convenience rather than a hard requirement. The authors' observation that modeling the score with a UNet might be more robust to misspecification than classical Markov random field algorithms is interesting. An interesting direction for future work could be to investigate what distributions beyond this class remain tractable for UNets.
> >
> > I have read the other reviews and responses as well. I maintain that this paper should be accepted. The restrictions in the theoretical analysis help simplify the exposition and could be relaxed in future work. I do not think that this paper requires experimental evaluations to be complete, as its message is (partly) supported by experimental evidence from the literature.

---

> > > ### Author Response · Authors · 2024-11-22
> > >
> > > We truly appreciate your support and your recognition of the paper's contributions. Thank you again for your valuable comments and for recommending acceptance!

---

### Official Review · Reviewer_f34u · 2024-11-05

**Soundness:** 3
**Presentation:** 3
**Contribution:** 3
**Rating:** 6
**Confidence:** 2

**Summary:**

U-Nets have shown promising performance across different computer vision tasks. In this paper, the authors aim to provide a theoretical explanation of the U-Net through the lens of belief propagation. Specifically, the authors have shown that the U-Net approximates the belief propagation denoising, which results in its specific model structure. In addition, the authors have shown that convolutional neural networks are well suited for classifications.

**Strengths:**

- Provide a rigorous theoretical explanation of U-Net and CNN through the lens of a graphical model

**Weaknesses:**

- Many hypotheses are made for the derivation. The conclusion would be strengthened if there were at least some toy-ish experiments to test these hypotheses
- (minor) Given how dense the paper is, a lookup table for notations would be very helpful

**Questions:**

- In Eq. 17 only ERM error is used while I think when these models are trained usually there's an extra regularization term, I was wondering does it influences the conclusion?

- In neural network, the upper bound in the theorem has the tendency to be too loose, I was wondering how tight are the bounds?

---

> ### Author Response · Authors · 2024-11-17
>
> We thank the reviewer for their thoughtful comments. Below, we address the identified weaknesses and questions.
>
> > "Many hypotheses are made for the derivation. The conclusion would be strengthened if there were at least some toy-ish experiments to test these hypotheses. "
>
> While we did not conduct numerical simulations, numerous experiments in the literature demonstrate the effectiveness of U-Nets in image-denoising tasks and hierarchical denoising models, including [KGMS23] and [SFW24]. We acknowledge the value of such experiments and regret that we may not be able to complete simulations in the short term. However, we plan to include experimental results in the final version.
>
> [KGMS23] Zahra Kadkhodaie, Florentin Guth, Stephane Mallat, and Eero P Simoncelli. Learning multi-scale local conditional probability models of images. ICLR 2023.
>
> [SFW24] Antonio Sclocchi, Alessandro Favero, and Matthieu Wyart. A phase transition in diffusion models reveals the hierarchical nature of data. arXiv preprint arXiv:2402.16991, 2024.
>
> > "In Eq. 17 only ERM error is used while I think when these models are trained usually there's an extra regularization term, I was wondering does it influences the conclusion?"
>
> Eq. (17) analyzes ERM over the neural network class with a norm constraint, which is critical for proving the generalization error bound. In practice, minimizing the norm-constrained ERM can be equivalently reformulated as minimizing the norm-regularized ERM, such as via SGD on the regularized empirical risk. Therefore, adding a regularization term does not significantly affect the conclusions drawn from the analysis.
>
> > In neural network, the upper bound in the theorem has the tendency to be too loose, I was wondering how tight are the bounds?
>
> We acknowledge that the upper bound in the theorem could be improved by adopting a more refined approximation scheme for the belief propagation algorithm. While the current bound may not be tight, it ensures that the required sample size scales polynomially with the dimension and relevant problem parameters, thereby avoiding the curse of dimensionality. This ability to circumvent the curse of dimensionality represents a key contribution of this paper.

---

> > ### Author Response · Authors · 2024-11-22
> >
> > We wanted to follow up on our previous response to your comments. We hope our clarifications address your concerns, and we’d be happy to answer your further questions.

---

> > ### Comment · Reviewer_f34u · 2024-11-25
> >
> > Thank you for the clarification and for addressing my concerns, I increased my confidence score.

---

### Official Review · Reviewer_iUQ6 · 2024-11-06

**Soundness:** 3
**Presentation:** 2
**Contribution:** 2
**Rating:** 6
**Confidence:** 2

**Summary:**

The paper examines the widely-used U-Net architecture under the scope of generative hierarchical models. The authors present a theoretical framework that treats the U-Net convolution and pooling layers as nodes of a hierarchical graphical model. Under their analysis, both classification and denoising with U-Nets can be seen as empirical risk minimization for a Bayes classifier/denoiser. The proposed interpretation can

**Strengths:**

- Interpreting U-Nets as hierarchical models is an interesting angle that could help explain some of the significant design choices behind training and inference in diffusion models.
- The writing of the paper is clear and helps with understanding the intuition behind the hierarchical model interpretation of U-Nets.

**Weaknesses:**

- The goal of the analysis is slightly unclear. The main result of the paper establishes a bound for the difference between the U-Net denoiser and the optimal denoiser, based on a simplified U-Net architecture. However, there is no motivation on why such a bound could be useful. See questions.

**Questions:**

- Why are the established bounds useful for future neural network architecture development? Could they help us understand scaling laws for U-Nets?
- What are the implications of interpreting each U-Net layer as a latent in the hierarchical model? The authors mention that "[...] our theoretical findings generated a hypothesis of the functionality of each layer of the U-Nets", but there are no actual hypotheses made. Can you conjecture something about the features learned in intermediate layers of the U-Net and how they have been used in other tasks [1]?

[1] Baranchuk, Dmitry, et al. "Label-Efficient Semantic Segmentation with Diffusion Models." International Conference on Learning Representations.

---

> ### Author Response · Authors · 2024-11-17
>
> We thank the reviewer for their thoughtful comments. Below, we address the identified weaknesses and questions.
> > ``The goal of the analysis is slightly unclear… However, there is no motivation on why such a bound could be useful. Why are the established bounds useful for future neural network architecture development? Could they help us understand scaling laws for U-Nets?’’
>
> The primary goal of this analysis is to address the challenge of the curse of dimensionality in neural network approximation theory: approximating general high-dimensional functions requires network sizes that scale exponentially with dimension $d$.
> Existing works mitigate the curse of dimensionality by leveraging smoothness or low-dimensional structure assumptions of the target function (e.g., [Bach17]). However, applying a smoothness assumption to the denoising function of image distributions imposes unnatural constraints on the density function in high-dimensional spaces, as noted in [OAS23].
>
> This work introduces a different approach by assuming a generative hierarchical model (GHM). This model generates high-level concepts first and progressively refines them into finer details, aligning with recent advances in modeling image and text distributions (e.g., [SFW24]). By leveraging this assumption, the neural network naturally approximates the belief propagation algorithm within the GHM framework, circumventing the curse of dimensionality.
>
> This framework offers a new statistical perspective on why U-Nets (and potentially other deep neural networks) perform effectively in denoising tasks and diffusion models. Understanding the broader implications of this theoretical framework is a key direction for future research, including its potential impact on architecture design and scaling laws.
>
> [OAS23] Kazusato Oko, Shunta Akiyama, and Taiji Suzuki, Diffusion models are minimax optimal distribution estimators, arXiv preprint arXiv:2303.01861 (2023).
>
> [Bach17] Francis Bach. Breaking the Curse of Dimensionality with Convex Neural Networks. JMLR 18(19):1−53, 2017.
>
> [SFW24] Antonio Sclocchi, Alessandro Favero, and Matthieu Wyart. A phase transition in diffusion models reveals the hierarchical nature of data. arXiv preprint arXiv:2402.16991, 2024.
>
> > ``What are the implications of interpreting each U-Net layer as a latent in the hierarchical model? The authors mention that "[...] our theoretical findings generated a hypothesis of the functionality of each layer of the U-Nets", but there are no actual hypotheses made. Can you conjecture something about the features learned in intermediate layers of the U-Net and how they have been used in other tasks [1]? ‘’
>
> Our approximation theory suggests that the intermediate layers of U-Nets correspond to the beliefs of latent nodes in the hierarchical model. Specifically: The top and bottom layers of U-Nets represent the most fine-grained features. The middle layers capture higher-level concepts, aligning with the hierarchical nature of the generative model.
>
> We propose the hypothesis that after training U-Nets via stochastic gradient algorithms, the intermediate layers represent the beliefs of latent nodes in the generative hierarchical model. This hypothesis aligns with empirical observations in [SFW24], which demonstrate the hierarchical nature of data in diffusion models. This perspective may also explain why the semantic information in U-Net's intermediate representations is valuable for tasks like segmentation, as supported by results in [1]. We plan to further investigate this hypothesis and its implications for downstream tasks in future work.
>
> [1] Baranchuk, Dmitry, et al. "Label-Efficient Semantic Segmentation with Diffusion Models." International Conference on Learning Representations.
>
> [SFW24] Antonio Sclocchi, Alessandro Favero, and Matthieu Wyart. A phase transition in diffusion models reveals the hierarchical nature of data. arXiv preprint arXiv:2402.16991, 2024.

---

> > ### Author Response · Authors · 2024-11-22
> >
> > We wanted to follow up on our previous response to your comments. We hope our clarifications address your concerns, and we’d be happy to answer your further questions.

---

> > > ### Comment · Reviewer_iUQ6 · 2024-11-28
> > >
> > > Thank you for spending the time to provide in-depth clarifications for my questions. I believe that this paper can be an important stepping stone to future analyses. I will be increasing my score to reflect my change in opinion.

---

### Meta-Review · Area_Chair_arQ2 · 2024-12-22

**Metareview:**

This work examines graphical models and the U-Net architecture in simplified form in order to theoretically relate the two. In particular, or a restricted setting, the U-Net architecture and computation can be connected with belief propagation and approximation bounds are derived. This is a purely theoretical paper, and relates to prior work in this direction, while providing novel insight into the design and working of diffusion models with U-Net architectures.

The main strengths and weaknesses are:

- Pro: Important problem and novel theoretical connection (UKfv: "studies an important problem [...] to break the curse of dimensionality when learning diffusion models")
- Pro: The theoretical content is insightful, new, and able to be extended further (f34u: "rigorous theoretical explanation of U-Net and CNN through the lens of a graphical model", UKfv: "The restrictions in the theoretical analysis help simplify the exposition and could be relaxed in future work")
- Pro: While there are no experiments in this work, this weakness is tempered by evidence from existing work (UKfv: "I do not think that this paper requires experimental evaluations to be complete, as its message is (partly) supported by experimental evidence from the literature."). Furthermore experiments are not required.
- Con: The theoretical content may not be valid w.r.t. practice because the operations studied do not reflect convolution as it is computed in deep networks (5xWe: "While it is helpful that the authors clarify these deviations from practical implementations, referring to these layers as “convolutional” may be misleading, as they do not fully adhere to the standard convolution operations")
- Con: No experiments to support the theoretical hypotheses and conclusions (f34u: "conclusion would be strengthened if there were at least some toy-ish experiments to test these hypotheses", xjiq: "it lacks any form of empirical validation of its conclusions").
- Con: Questions about the purpose and value of the analysis and what precisely the interpretation of features as beliefs of latent nodes implies (iUQ6: "[...] there is no motivation on why such a bound could be useful.").

The strongest arguments for acceptance are the theoretical connection between deep nets and graphical models (in the given restricted scope) and the importance of the topic given the popularity of diffusion. While the connection is not complete, in that it does not extend to deep nets as they are used in practice, it has been judged as theoretically sound and can inform future work. Furthermore the work is well-connected to the recent literature, and carries on from the previous steps in this direction.

The strongest arguments for rejection are 1. the disconnect from networks studied and deep networks (specifically: U-Nets and ConvNets) as they are usually defined and used 2. the lack of experiments to justify theoretical hypotheses and check conclusions given the distance between the theory and practice as already noted.

Five expert reviewers agree on acceptance, although 4/5 of the reviewers vote only for marginal accept (rating: 6). However the reviewer UKfv votes for clear acceptance (rating: 8). UKfv is the most confident, has the most related background, and defends the paper during the discussion. The meta-reviewer sides with acceptance, though cautions the authors to include experiments and a discussion of the gaps between theory and practice in the final revision in order to achieve greater impact. This suggestion is in line with the authors' own statement that "we plan to include experimental results in the final version". This is merely a suggestion to emphasize the comments of multiple reviewers, and not a condition on acceptance, as the submission is sufficiently informative theoretically for publication per the evaluation of the reviewers.

**Additional Comments On Reviewer Discussion:**

Thea authors provide a response to each review and every reviewer has responded in turn. UKfv maintains their rating of 8 and argues for acceptance. iUQ6 increases their rating from 5 to 6. f34u increased their confidence from 1 to 2. Following the author-reviewer discussion no AC-reviewer discussion was needed as every author-reviewer thread had already converged with a clear result. The points about experiments (f3ru, xjiq, UKfv) and in the relevance and significance of the theoretical contributions (5xWe, UKfv) were the most salient. These were addressed by the rebuttal to the reviewers' satisfaction insofar as ratings and confidence were maintained or improved as confirmed by their concluding comments.

---

### Decision · Program_Chairs · 2025-01-22

Accept (Poster)